# On the Importance of the Policy Structure in Offline Reinforcement Learning

## Abstract

Offline reinforcement learning (RL) has attracted a great deal of attention recently as an approach to utilizing past experience to learn a policy. Recent studies have reported the challenges of offline RL, such as estimating the values of actions that are out of the data distribution. To mitigate the issues of offline RL, we propose an algorithm that leverages a mixture of deterministic policies. With our framework, the state-action space is divided by learning discrete latent variables, and sub-policies corresponding to each region are trained. The proposed algorithm, which we call Value-Weighted Variational Auto-Encoder (V2AE), is derived by considering the variational lower bound of the offline RL objective function. The aim of this work is to shed light on the importance of the policy structure in offline RL. We show empirically that the use of the proposed mixture policy can reduce the accumulation of the critic loss in offline RL, which was reported in previous studies. Experimental results also indicate that introducing the policy structure improves the performance on tasks with D4RL benchmarking datasets.

## 1 Introduction

Reinforcement learning (RL) (Sutton & Barto, 2018) has had remarkable success in a variety of applications. Many of its successes have been achieved in online learning settings where the RL agent interacts with the environment during the learning process. However, such interactions are often time consuming and computationally expensive. The desirability of reducing the number of interactions in RL has motivated an active interest in offline RL (Levine et al., 2020), also known as batch RL (Lange et al., 2012). In offline RL, the goal is to learn the optimal policy from a prepared dataset collected through an arbitrary and unknown process. Prior work on offline RL has focused on how to avoid estimating the Q-values of action that are out of the data distribution (Fujimoto et al., 2019; Fujimoto & Gu, 2021). While previous studies often address this issue in terms of the regularization of critics (Kumar et al., 2020; An et al., 2021; Kostrikov et al., 2021; 2022), we propose to mitigate the issue from the perspective of the policy structure. Our hypothesis is that evaluation of the out-of-distribution actions can be avoided by dividing the state-action space, which is potentially achieved by learning discrete latent variables of the state-action space. When the data distribution is multimodal, as shown in Figure 1(a), fitting a policy modeled with a unimodal distribution such as a Gaussian distribution may lead to interpolation between separate modes, which will result in the value estimation of actions that are out of the data distribution (Figure 1(b)). To avoid this, we employ a mixture of deterministic policies (Figure 1(c)). We divide the state-action space and learn sub-policies for each region. Ideally, this approach will enable us to avoid interpolating separate modes of the data distribution.

In this study, we propose to train a mixture policy by learning discrete latent representations, which can be interpreted as dividing the state-action space and learning sub-policies that correspond to each region. We derive the proposed algorithm by considering the variational lower bound of the offline RL objective function. We refer to the proposed algorithm as Value-Weighted Variational Auto-Encoder (V2AE). The main contribution of this study is an offline RL algorithm that trains a mixture policy by learning discrete latent variables. We also propose a regularization technique for a mixture policy based on the mutual information. We empirically show that the proposed regularization technique improves the performance of the proposed algorithm. A previous study in (Brandfonbrener et al., 2021) reports the accumulation of the critic loss values during the training phase, which was considered the result of generating out-of-distribution actions. We show empiri-

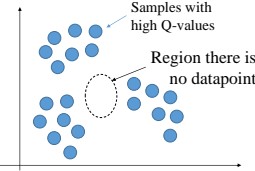

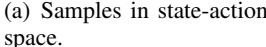

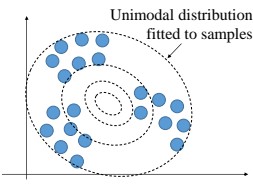

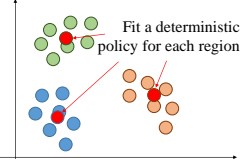

(a) Samples in state-action space.

(b) Result of fitting a unimodal distribution.

(c) Proposed approach.

Figure 1: Schematic illustration of the proposed approach. (a) In offline RL, the distribution of samples is often multimodal; (b) Fitting a unimodal distribution to such samples can lead to estimating the action out of the data distribution; (c) In the proposed approach, the latent discrete variable of the state-action space is learned, and a deterministic policy is learned for each region.

cally that the use of the proposed mixture policy can reduce the accumulation of the approximation error in offline RL. In experiments with benchmark tasks in D4RL (Fu et al., 2020), the proposed algorithms proved to be competitive with the popular offline RL methods. While the experimental result shows the promising performance , we aim to shed light on the importance of the policy structure as inductive bias in offline RL, rather than claim the state-of-the-art performance.

## 2 RELATED WORK

Recent studies have shown that regularization is the crucial component for offline RL (Fujimoto et al., 2019; Kumar et al., 2020; Levine et al., 2020; Kostrikov et al., 2021). For example, Kostrikov et al. (2021) proposed a regularization based on Fischer divergence, and Fujimoto & Gu (2021) showed that simply adding a behavior cloning term to the objective function in TD3 can achieve state-of-the-art performance on D4RL benchmark tasks (Fu et al., 2020). Other research has investigated the structure of the critic, proposing the use of an ensemble of critics (An et al., 2021) or offering a one-step offline RL approach (Brandfonbrener et al., 2021; Goo & Niekum, 2021). Previous studies (Fujimoto et al., 2019; Fujimoto & Gu, 2021) have indicated that the source of the value approximation error is "extrapolation error" that occurs when the value of state-action pairs that are not contained in a given dataset is estimated. Our hypothesis is that such "extrapolation error" can be mitigated by dividing the state-action space, which is potentially achieved by learning discrete latent variables. We investigate the effect of incorporating the policy structure as inductive bias in offline RL, which has not been fully investigated.

Learning the discrete latent variable in the context of RL is closely related to a mixture policy, where a policy is represented as a combination of a finite number of sub-policies. In a mixture policy, one of the sub-policies is activated for a given state, and the module that determines which sub-policy to use is often called the gating policy (Daniel et al., 2016). Because of the two-layered structure, a mixture policy is also called *a hierarchical policy* (Daniel et al., 2016). Although we do not consider temporal abstraction in this study, we note that a well-known hierarchical RL framework with temporal abstraction is the option critic (Bacon et al., 2017). Since we consider policies without temporal abstraction, we use the term "mixture policy," following the terminology in Wulfmeier et al. (2021). Previous studies have demonstrated the advantages of mixture policies in online RL (Osa et al., 2019; Zhang & Whiteson, 2019; Wulfmeier et al., 2020; 2021; Akrour et al., 2021). In these existing methods, sub-policies are often trained to cover separate modes of the Q-function, which is similar to our idea. While existing methods have leveraged the latent variable in offline RL (Zhou et al., 2020; Chen et al., 2021b; 2022), the latent variable is continuous in these methods. As indicated by studies on latent representations (Kingma & Welling, 2014; Dupont, 2018; Brown et al., 2020), we think that the use of the discrete latent variable should be investigated in offline RL.

## 3 PROBLEM FORMULATION

**Reinforcement Learning**  Consider a reinforcement learning problem under a Markov decision process (MDP) defined by a tuple $(\mathcal{S}, \mathcal{A}, \mathcal{P}, r, \gamma, d)$, where $\mathcal{S}$ is the state space, $\mathcal{A}$ is the action space, $\mathcal{P}(\boldsymbol{s}_{t+1}|\boldsymbol{s}_t, \boldsymbol{a}_t)$ is the transition probability density, $r(\boldsymbol{s}, \boldsymbol{a})$ is the reward function, $\gamma$ is the discount

factor, and $d(\boldsymbol{s}_0)$ is the probability density of the initial state. A policy $\pi(\boldsymbol{a}|\boldsymbol{s}) : \mathcal{S} \times \mathcal{A} \mapsto \mathbb{R}$ is defined as the conditional probability density over actions given states. The goal of RL is to identify a policy that maximizes the expected return $\mathbb{E}[R_0|\pi]$, where the return is the sum of discounted rewards over time given by $R_t = \sum_{k=t}^{T} \gamma^{k-t} r(\boldsymbol{s}_k, \boldsymbol{a}_k)$. The Q-function, $Q^\pi(\boldsymbol{s}, \boldsymbol{a})$, is defined as the expected return when starting from state $\boldsymbol{s}$ and taking action $\boldsymbol{a}$, then following policy $\pi$ under a given MDP (Sutton & Barto, 2018).

In offline RL, it is assumed that the learning agent is given a fixed dataset, $\mathcal{D} = \{(\boldsymbol{s}_i, \boldsymbol{a}_i, r_i)\}_{i=1}^N$, that consists of states, actions, and rewards collected by an unknown behavior policy. The goal of offline RL is to obtain a policy that maximizes the expected return using $\mathcal{D}$, without online interactions with an environment during the learning process.

**Objective function**   We formulate the problem of offline RL as follows. Given a dataset $\mathcal{D} = \{(\boldsymbol{s}_i, \boldsymbol{a}_i, r_i)\}_{i=1}^N$ obtained through the interactions between a behavior policy $\beta(\boldsymbol{a}|\boldsymbol{s})$ and the environment, our goal is to obtain a policy $\pi$ that maximizes the expected return. In the process of training a policy in offline RL, the expected return is evaluated with respect to the states stored in the given dataset. Thus, the objective function is given by

$$J(\pi) = \mathbb{E}_{\boldsymbol{s} \sim \mathcal{D}, \boldsymbol{a} \sim \pi} \left[ f^\pi(\boldsymbol{s}, \boldsymbol{a}) \right], \tag{1}$$

where $f^\pi$ is a function that quantifies the performance of policy $\pi$. In RL, there are several choices for $f^\pi$ as indicated in Schulman et al. (2016). TD3 employed the action-value function as $f^\pi(\boldsymbol{s}, \boldsymbol{a}) = Q^\pi(\boldsymbol{s}, \boldsymbol{a})$, and A2C employed the advantage-function as $f^\pi(\boldsymbol{s}, \boldsymbol{a}) = A^\pi(\boldsymbol{s}, \boldsymbol{a})$ (Mnih et al., 2016). Other previous studies employ shaping with the exponential function as $f^\pi(\boldsymbol{s}, \boldsymbol{a}) = \exp\left(Q^\pi(\boldsymbol{s}, \boldsymbol{a})\right)$ (Peters & Schaal, 2007) or $f^\pi(\boldsymbol{s}, \boldsymbol{a}) = \exp\left(A^\pi(\boldsymbol{s}, \boldsymbol{a})\right)$ (Neumann & Peters, 2008; Wang et al., 2018). Without loss of generality, we assume that the objective function is given by equation 1. We derive the proposed algorithm by considering the lower bound of the objective function of offline RL in equation 1.

**Mixture policy**   In this study, we consider a mixture of policy given by

$$\pi(\boldsymbol{a}|\boldsymbol{s}) = \sum_{\boldsymbol{z} \in \mathcal{Z}} \pi_{\text{gate}}(\boldsymbol{z}|\boldsymbol{s}) \pi_{\text{sub}}(\boldsymbol{a}|\boldsymbol{s}, \boldsymbol{z}), \tag{2}$$

where $\boldsymbol{z}$ is a discrete latent variable, $\pi_{\text{gate}}(\boldsymbol{z}|\boldsymbol{s})$ is the gating policy that determines the value of the latent variable, and $\pi_{\text{sub}}(\boldsymbol{a}|\boldsymbol{s}, \boldsymbol{z})$ is the sub-policy that determines the action for given $\boldsymbol{s}$ and $\boldsymbol{z}$. We assume that a sub-policy $\pi_{\text{sub}}(\boldsymbol{a}|\boldsymbol{s}, \boldsymbol{z})$ is deterministic; the sub-policy determines the action for given $\boldsymbol{s}$ and $\boldsymbol{z}$ in a deterministic manner as $\boldsymbol{a} = \boldsymbol{\mu_\theta}(\boldsymbol{s}, \boldsymbol{z})$, where $\boldsymbol{\mu_\theta}(\boldsymbol{s}, \boldsymbol{z})$ is parameterized with a vector $\boldsymbol{\theta}$. Additionally, we assume that the gating policy $\pi_{\text{gate}}(\boldsymbol{z}|\boldsymbol{s})$ determines the latent variable as

$$\boldsymbol{z} = \arg\max_{\boldsymbol{z}'} Q_{\boldsymbol{w}}(\boldsymbol{s}, \boldsymbol{\mu_\theta}(\boldsymbol{s}, \boldsymbol{z}')), \tag{3}$$

where $Q_{\boldsymbol{w}}(\boldsymbol{s}, \boldsymbol{a})$ is the estimated Q-function parameterized with a vector $\boldsymbol{w}$. This gating policy is applicable to the objective functions such as $f^\pi(\boldsymbol{s}, \boldsymbol{a}) = \exp\left(Q^\pi(\boldsymbol{s}, \boldsymbol{a})\right)$, $f^\pi(\boldsymbol{s}, \boldsymbol{a}) = A^\pi(\boldsymbol{s}, \boldsymbol{a})$, and $f^\pi(\boldsymbol{s}, \boldsymbol{a}) = \exp\left(A^\pi(\boldsymbol{s}, \boldsymbol{a})\right)$. Please refer to Appendix A for details.

## 4   TRAINING A MIXTURE POLICY BY MAXIMIZING THE VARIATIONAL LOWER BOUND

We consider a training procedure based on policy iteration (Sutton & Barto, 2018), where the critic and the policy are iteratively improved. In this section, we describe the policy update procedure in the proposed method.

To derive the update rule for the policy parameter $\boldsymbol{\theta}$, we first consider the lower bound of the objective function $\log J(\pi)$ in equation 1. We assume that $f^\pi(\boldsymbol{s}, \boldsymbol{a})$ in equation 1 is approximated with $\hat{f}_{\boldsymbol{w}}^\pi(\boldsymbol{s}, \boldsymbol{a})$, which is parameterized with a vector $\boldsymbol{w}$. In a manner similar to Dayan & Hinton (1997); Kober & Peters (2011), when $\hat{f}_{\boldsymbol{w}}^\pi(\boldsymbol{s}, \boldsymbol{a}) > 0$ for any $\boldsymbol{s}$ and $\boldsymbol{a}$, we can determine the lower bound of

$\log J(\pi)$ using Jensen's inequality as follows:

$$\log J(\pi) \approx \log \int d^\beta(\boldsymbol{s}) \pi_{\boldsymbol{\theta}}(\boldsymbol{a}|\boldsymbol{s}) \hat{f}_{\boldsymbol{w}}^\pi(\boldsymbol{s}, \boldsymbol{a}) \mathrm{d}\boldsymbol{s} \mathrm{d}\boldsymbol{a} \tag{4}$$

$$= \log \int d^\beta(\boldsymbol{s}) \beta(\boldsymbol{a}|\boldsymbol{s}) \frac{\pi_{\boldsymbol{\theta}}(\boldsymbol{a}|\boldsymbol{s})}{\beta(\boldsymbol{a}|\boldsymbol{s})} \hat{f}_{\boldsymbol{w}}^\pi(\boldsymbol{s}, \boldsymbol{a}) \mathrm{d}\boldsymbol{s} \mathrm{d}\boldsymbol{a} \geq \int d^\beta(\boldsymbol{s}) \beta(\boldsymbol{a}|\boldsymbol{s}) \log \frac{\pi_{\boldsymbol{\theta}}(\boldsymbol{a}|\boldsymbol{s})}{\beta(\boldsymbol{a}|\boldsymbol{s})} \hat{f}_{\boldsymbol{w}}^\pi(\boldsymbol{s}, \boldsymbol{a}) \mathrm{d}\boldsymbol{s} \mathrm{d}\boldsymbol{a} \tag{5}$$

$$= \mathbb{E}_{(\boldsymbol{s},\boldsymbol{a})\sim\mathcal{D}} \left[ \log \pi_{\boldsymbol{\theta}}(\boldsymbol{a}|\boldsymbol{s}) \hat{f}_{\boldsymbol{w}}^\pi(\boldsymbol{s}, \boldsymbol{a}) \right] - \mathbb{E}_{(\boldsymbol{s},\boldsymbol{a})\sim\mathcal{D}} \left[ \log \beta(\boldsymbol{a}|\boldsymbol{s}) \hat{f}_{\boldsymbol{w}}^\pi(\boldsymbol{s}, \boldsymbol{a}) \right], \tag{6}$$

where $\beta(\boldsymbol{a}|\boldsymbol{s})$ is the behavior policy, which is used for collecting the given dataset, and $d^\beta(\boldsymbol{s})$ is the stationary distribution over the state induced by executing the behavior policy $\beta(\boldsymbol{a}|\boldsymbol{s})$. The second term in equation 6 is independent of the policy parameter $\boldsymbol{\theta}$. Thus, we can maximize the lower bound of $J(\pi)$ by maximizing $\sum_{i=1}^N \log \pi_{\boldsymbol{\theta}}(\boldsymbol{a}_i|\boldsymbol{s}_i) \hat{f}_{\boldsymbol{w}}^\pi(\boldsymbol{s}_i, \boldsymbol{a}_i)$. When we employ $f^\pi(\boldsymbol{s}, \boldsymbol{a}) = \exp(A^\pi(\boldsymbol{s}, \boldsymbol{a}))$ and the policy is Gaussian, the resulting algorithm is equivalent to AWAC (Nair et al., 2020). To employ a mixture policy with a discrete latent variable, we further analyze the objective function in equation 6. As in Kingma & Welling (2014); Sohn et al. (2015), we can obtain a variant of the variational lower bound of the conditional log-likelihood:

$$\log \pi_{\boldsymbol{\theta}}(\boldsymbol{a}_i|\boldsymbol{s}_i) \geq -D_{\mathrm{KL}}(q_{\boldsymbol{\phi}}(\boldsymbol{z}|\boldsymbol{s}_i, \boldsymbol{a}_i)||p(\boldsymbol{z}|\boldsymbol{s}_i)) + \mathbb{E}_{\boldsymbol{z}\sim p(\boldsymbol{z}|\boldsymbol{s}_i, \boldsymbol{a}_i)} [\log \pi_{\boldsymbol{\theta}}(\boldsymbol{a}_i|\boldsymbol{s}_i, \boldsymbol{z})]$$
$$= \ell_{\mathrm{cvae}}(\boldsymbol{s}_i, \boldsymbol{a}_i; \boldsymbol{\theta}, \boldsymbol{\phi}), \tag{7}$$

where $q_{\boldsymbol{\phi}}(\boldsymbol{z}|\boldsymbol{s}, \boldsymbol{a})$ is the approximate posterior distribution parameterized with a vector $\boldsymbol{\phi}$, and $p(\boldsymbol{z}|\boldsymbol{s})$ is the true posterior distribution. The derivation of equation 7 is provided in Appendix B. Although it is often assumed in prior studies (Fujimoto et al., 2019) that $\boldsymbol{z}$ is statistically independent of $\boldsymbol{s}$, i.e., $p(\boldsymbol{z}|\boldsymbol{s}) = p(\boldsymbol{z})$, in our framework $p(\boldsymbol{z}|\boldsymbol{s})$ should represent the behavior of the gating policy $\pi_{\boldsymbol{\theta}}(\boldsymbol{z}|\boldsymbol{s})$. Recognizing the challenge of representing exactly the gating policy $\pi_{\boldsymbol{\theta}}(\boldsymbol{z}|\boldsymbol{s})$ in equation 3, we approximate it with the softmax distribution given by

$$p(\boldsymbol{z}|\boldsymbol{s}) = \frac{\exp\left(Q_{\boldsymbol{w}}(\boldsymbol{s}, \boldsymbol{\mu}_{\boldsymbol{\theta}}(\boldsymbol{s}, \boldsymbol{z}))\right)}{\sum_{\boldsymbol{z}\in\mathcal{Z}} \exp\left(Q_{\boldsymbol{w}}(\boldsymbol{s}, \boldsymbol{\mu}_{\boldsymbol{\theta}}(\boldsymbol{s}, \boldsymbol{z}))\right)}. \tag{8}$$

Since we employ double-clipped Q-learning as in Fujimoto et al. (2018), we compute $Q_{\boldsymbol{w}}(\boldsymbol{s}, \boldsymbol{\mu}_{\boldsymbol{\theta}}(\boldsymbol{s}, \boldsymbol{z})) = \min_{j=1,2} Q_{\boldsymbol{w}_j}(\boldsymbol{s}, \boldsymbol{\mu}_{\boldsymbol{\theta}}(\boldsymbol{s}, \boldsymbol{z}))$ in our implementation. The second term in equation 7 is approximated as the mean squared error, as in the standard implementation of VAE. Based on equation 6 and equation 7, we obtain the objective function for training a mixture policy as

$$\mathcal{L}_{\mathrm{ML}}(\boldsymbol{\theta}, \boldsymbol{\phi}) = \sum_{i=1}^N f^\pi(\boldsymbol{s}_i, \boldsymbol{a}_i) \ell_{\mathrm{cvae}}(\boldsymbol{s}_i, \boldsymbol{a}_i; \boldsymbol{\theta}, \boldsymbol{\phi}). \tag{9}$$

This objective can be regarded as the weighted maximum likelihood (Kober & Peters, 2011) for a mixture policy. Our objective function can be viewed as reconstructing the state-action action pairs with adaptive weights, as in Peters & Schaal (2007); Nair et al. (2020). Therefore, the policy samples actions within the support and does not evaluate out-of-distribution actions. The primary difference between the proposed method and the existing methods (Peters & Schaal, 2007; Nair et al., 2020) is that the use of a mixture of policies conditioned on discrete latent variables in our approach can be regarded as dividing the state-action space. For example, in AWAC (Nair et al., 2020), a unimodal policy is used to reconstruct all of the "good" actions in the given dataset. However, in the context of offline RL, the given dataset may contain samples collected by diverse behaviors, and enforcing the policy to cover all the modes in the dataset can degrade the resulting performance. In our approach, the policy $\pi_{\boldsymbol{\theta}}(\boldsymbol{a}|\boldsymbol{s}, \boldsymbol{z})$ is encouraged to mimic the state-action pairs which are assigned to the same values of $\boldsymbol{z}$, without mimicking the actions which are assigned the different values of $\boldsymbol{z}$.

In addition, we also propose a regularization technique for a mixture policy based on the mutual information (MI) between $\boldsymbol{z}$ and the state action pair $(\boldsymbol{s}, \boldsymbol{a})$, which we denote by $I(\boldsymbol{z}; \boldsymbol{s}, \boldsymbol{a})$. As shown by Barber & Agakov (2003), the variational lower bound of $I(\boldsymbol{z}; \boldsymbol{s}, \boldsymbol{a})$ is given by $\mathbb{E}[\log g_{\boldsymbol{\psi}}(\boldsymbol{z}|\boldsymbol{s}, \boldsymbol{a})]$, where $g_{\boldsymbol{\psi}}(\boldsymbol{z}|\boldsymbol{s}, \boldsymbol{a})$ is an auxiliary distribution to approximate the posterior distribution $p(\boldsymbol{z}|\boldsymbol{s}, \boldsymbol{a})$. Thus, the final objective function is given by

$$\mathcal{L}(\boldsymbol{\theta}, \boldsymbol{\phi}, \boldsymbol{\psi}) = \mathcal{L}_{\mathrm{ML}}(\boldsymbol{\theta}, \boldsymbol{\phi}) + \lambda \sum_{i=1}^N \mathbb{E}_{\boldsymbol{z}\sim p(\boldsymbol{z})} \log g_{\boldsymbol{\psi}}(\boldsymbol{z}|\boldsymbol{s}_i, \boldsymbol{\mu}_{\boldsymbol{\theta}}(\boldsymbol{s}_i, \boldsymbol{z})). \tag{10}$$

The MI-based regularization using the second term in equation 10 encourages the diversity of the behaviors encoded in the sub-policy $\pi(\boldsymbol{a}|\boldsymbol{s}, \boldsymbol{z})$. We empirically show that this regularization improves the performance of the proposed method in Section 7.

---

**Algorithm 1** Value-Weighted Variational Auto-Encoder (V2AE)

Initialize the actor $\boldsymbol{\mu_\theta}$, critic $Q_{\boldsymbol{w}_j}$ for $j = 1, 2$, and the posterior $q_\phi(\boldsymbol{z}|\boldsymbol{s}, \boldsymbol{a})$
**for** $t = 1$ **to** $T$ **do**
    Sample a minibatch $\{(\boldsymbol{s}_i, \boldsymbol{a}_i, \boldsymbol{s}'_i, r_i)\}_{i=1}^M$ from $\mathcal{D}$
    **for** each element $(\boldsymbol{s}_i, \boldsymbol{a}_i, \boldsymbol{s}'_i, r_i)$ **do**
        Compute the value of the latent variable for $\boldsymbol{s}'$: $\boldsymbol{z}' = \arg\max_{\tilde{\boldsymbol{z}}'} Q_{\boldsymbol{w}}(\boldsymbol{s}', \boldsymbol{\mu_{\theta'}}(\boldsymbol{s}', \tilde{\boldsymbol{z}}'))$
        Compute the target value: $y_i = r + \gamma \min_{j=1,2} Q_{\boldsymbol{w}_j}(\boldsymbol{s}, \boldsymbol{\mu_{\theta'}}(\boldsymbol{s}', \boldsymbol{z}'))$
    **end for**
    Update the critic by minimizing $\sum_{i=1}^M \left\| y_i - Q_{\boldsymbol{w}_j}(\boldsymbol{s}_i, \boldsymbol{a}_i) \right\|^2$ for $j = 1, 2$
    **if** $t \mod d_{\text{interval}} = 0$ **then**
        Update the actor and the posterior by maximizing equation 9
        (optionally) Update the actor by maximizing $\sum_{i=1}^M \mathbb{E}_{\boldsymbol{z} \sim p(\boldsymbol{z})} \log g_\psi(\boldsymbol{z}|\boldsymbol{s}_i, \boldsymbol{\mu_\theta}(\boldsymbol{s}_i, \boldsymbol{z}))$
    **end if**
**end for**

---

## 5    TRAINING THE CRITIC FOR A MIXTURE OF DETERMINISTIC POLICIES

To derive the objective function for training the critic for the mixture of deterministic policies using the gating policy in equation 3, we consider the following operator:

$$\mathcal{T}_{\boldsymbol{z}} Q_{\boldsymbol{z}} = r(\boldsymbol{s}, \boldsymbol{a}) + \gamma \mathbb{E}_{\boldsymbol{s}'} \left[ \max_{\boldsymbol{z}'} Q_{\boldsymbol{z}}(\boldsymbol{s}', \boldsymbol{\mu}(\boldsymbol{s}', \boldsymbol{z}')) \right]. \tag{11}$$

We refer to the operator $\mathcal{T}_{\boldsymbol{z}}$ as the *latent-max-Q operator*. Following Ghasemipour et al. (2021), we prove the following theorems.

**Theorem 5.1.** *In the tabular setting, $\mathcal{T}_{\boldsymbol{z}}$ is a contraction operator in the $\mathcal{L}_\infty$ norm. Hence, with repeated applications of the $\mathcal{T}_{\boldsymbol{z}}$, any initial Q function converges to a unique fixed point.*

The proof of Theorem 5.1 is provided in the Appendix C.

**Theorem 5.2.** *Let $Q_{\boldsymbol{z}}$ denote the unique fixed point achieved in Theorem 5.1, and let $\pi_{\boldsymbol{z}}$ denote the policy that chooses the latent variable as $\boldsymbol{z} = \arg\max_{\boldsymbol{z}'} Q(\boldsymbol{s}, \boldsymbol{\mu}(\boldsymbol{s}, \boldsymbol{z}'))$ and outputs the action given by $\boldsymbol{\mu}(\boldsymbol{s}, \boldsymbol{z})$ in a deterministic manner. Then $Q_{\boldsymbol{z}}$ is the Q-value function corresponding to $\pi_{\boldsymbol{z}}$.*

*Proof.* (Theorem 5.2) Rearranging equation 11 with $\boldsymbol{z}' = \arg\max Q_{\boldsymbol{z}}(\boldsymbol{s}', \boldsymbol{\mu}(\boldsymbol{s}', \boldsymbol{z}'))$, we obtain

$$\mathcal{T}_{\boldsymbol{z}} Q_{\boldsymbol{z}} = r(\boldsymbol{s}, \boldsymbol{a}) + \gamma \mathbb{E}_{\boldsymbol{s}'} \mathbb{E}_{\boldsymbol{a}' \sim \pi_{\boldsymbol{z}}} \left[ Q_{\boldsymbol{z}}(\boldsymbol{s}', \boldsymbol{a}') \right]. \tag{12}$$

Since by definition $Q_{\boldsymbol{z}}$ is the unique fixed point of $\mathcal{T}_{\boldsymbol{z}}$, we have our result. $\square$

These theorems show that the latent-max-Q operator $\mathcal{T}_{\boldsymbol{z}}$ retains the contraction and fixed-point existence properties. Based on these results, we estimate the Q-function by applying the latent-max-Q operator. In our implementation, we employed double-clipped Q-learning (Fujimoto et al., 2018). Thus, given a dataset $\mathcal{D}$, the critic is trained by minimizing

$$\mathcal{L}(\boldsymbol{w}_j) = \sum_{(\boldsymbol{s}_i, \boldsymbol{a}_i, \boldsymbol{s}'_i, r_i) \in \mathcal{D}} \left\| Q_{\boldsymbol{w}_j}(\boldsymbol{s}_i, \boldsymbol{a}_i) - y_i \right\|^2 \tag{13}$$

for $j = 1, 2$, where the target value $y_i$ is computed as

$$y_i = r_i + \gamma \max_{\boldsymbol{z}' \in \mathcal{Z}} \min_{j=1,2} Q_{\boldsymbol{w}_j}(\boldsymbol{s}', \boldsymbol{\mu_{\theta'}}(\boldsymbol{s}'_i, \boldsymbol{z}')). \tag{14}$$

## 6    PRACTICAL IMPLEMENTATION

The proposed Value-Weighted Variational Auto-Encoder (V2AE) algorithm is summarized in Algorithm 1. As in TD3 (Fujimoto et al., 2018), the actor is updated once every after $d_{\text{interval}}$ updates of the critics. In our implementation, we set $d_{\text{interval}} = 2$. For modeling the discrete latent variable, we use the Gumbel-softmax trick (Jang et al., 2017; Maddison et al., 2017). We also use the state

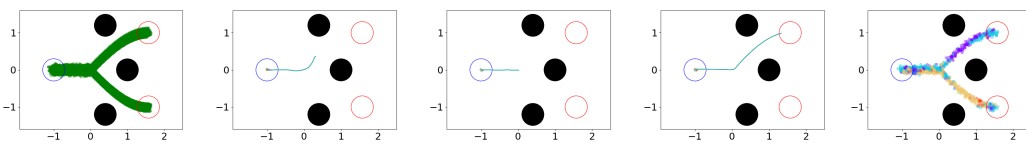

(a) Task setting and data samples.
(b) Behavior obtained by TD3+BC.
(c) Behavior obtained by AWAC.
(d) Behavior obtained by V2AE.
(e) Visualization of the latent variable in V2AE.

Figure 2: Performance on a simple task with multimodal data distribution.

Table 1: Algorithm setup in the experiment.

|  | TD3+BC | AWAC | V2AE |
|---|---|---|---|
| Critic training | double-clipped Q-learning | double-clipped Q-learning | double-clipped Q-learning |
| Policy type | monolithic & deterministic | monolithic & stochastic | mixture & deterministic |
| Regularization | BC term | none | none |
| State | normalized | normalized | normalized |

normalization used in TD3+BC (Fujimoto & Gu, 2021). In preliminary experiments, we found that when $f^\pi(s, a) = \exp(bA^\pi(s, a))$ in equation 9, the scaling factor $b$ has non-trivial effects on performance and the best value of $b$ is different for each task. To avoid changing the scaling parameter for each task, we used the normalization of the advantage function as

$$f^\pi(s, a) = \exp\left(\frac{\alpha\left(A^\pi(s, a) - \max_{(\tilde{s},\tilde{a})\in\mathcal{D}_{\text{batch}}} A^\pi(\tilde{s}, \tilde{a})\right)}{\max_{(\tilde{s},\tilde{a})\in\mathcal{D}_{\text{batch}}} A^\pi(\tilde{s}, \tilde{a}) - \min_{(\tilde{s},\tilde{a})\in\mathcal{D}_{\text{batch}}} A^\pi(\tilde{s}, \tilde{a})}\right), \quad (15)$$

where $\mathcal{D}_{\text{batch}}$ is a mini-batch sampled from the given dataset $\mathcal{D}$ and $\alpha$ is a constant; we set $\alpha = 10$ for Mujoco tasks in our experiments. For other hyperparameter details, please refer to the Appendix L.

# 7 EXPERIMENTS

We investigated the effect of the policy structure on the resulting performance and the training error of the critics. In the first experiment, we performed a comparative assessment of TD3+BC (Fujimoto & Gu, 2021), AWAC (Nair et al., 2020) and V2AE on a toy problem, where the distribution of samples in a given dataset is multimodal. We also conducted a quantitative comparison between the proposed methods and baseline methods with D4RL benchmark tasks (Fu et al., 2020). In the following section, we refer to the proposed method based on the objective in equation 9 as V2AE and a variant of the proposed method with the MI-based regularization in equation 10 as infoV2AE. In both the toy problem and the D4RL tasks, we used the author-provided implementations of TD3+BC, and our implementation of V2AE and AWAC are based on the author-implementation of TD3+BC.

## 7.1 MULTIMODAL DATA DISTRIBUTION ON TOY TASK

To show the effect of the multimodal data distribution in a given dataset, we evaluated the performance of TD3+BC, AWAC, and V2AE on a toy task shown in Figure 2. The differences between the compared methods are summarized in Table 1. In our implementation of AWAC and V2AE, we used the state normalization and double-clipped Q-learning as in TD3+BC and the normalization of the advantage function described in Section 6. The difference between AWAC and V2AE indicates the effect of the policy representation. In this toy task, the agent is represented as a point mass, the state is the position of the point mass in two-dimensional space, and the action is the small displacement of the point mass. There are two goals in this task, which are indicated by red circles in Figure 2. A blue circle indicates the start position in Figure 2, and there are three obstacles, which are indicated by black solid circles. In this task, the reward is sparse: When the agent reaches one of the goals, the agent receives the reward 1.0, and the episode ends. If the agent makes a contact with one of the obstacles, the agent receives the reward -1.0, and the episode ends. In the given dataset, trajectories to the two goals are provided, and there is no information on which goal the agent is heading.

The score is summarized in Table 2. Among the evaluated methods, only V2AE successfully solved the task. The policy obtained by TD3+BC did not reach the goal in a

Table 2: Performance on the toy task.

| TD3+BC | AWAC | V2AE |
|---|---|---|
| -0.2± 0.4 | 0.33± 0.44 | **1.0±0.0** |

Table 3: Results on mujoco tasks using D4RL-v2 datasets and AntMaze tasks. Average normalized scores over the last 10 test episodes and five seeds are shown. HC = HalfCheetah, HP = Hopper, WK = Walker2d.

| | | TD3+BC (re-run) | CQL (re-run) | easyBCQ (re-run) | AWAC (ours) | V2AE (ours) | infoV2AE (ours) |
|---|---|---|---|---|---|---|---|
| Expert | HC | **96.3±0.9** | 22.0±9.6 | 95.2± 1.8 | 94.8± 0.2 | **97.0± 1.0** | 95.6± 2.0 |
| | HP | **109.9±2.5** | 105.8±3.8 | 70.4± 8.0 | 109.8± 2.9 | 93.6± 15.1 | 107.5± 2.9 |
| | WK | 110.2±0.4 | 108.9±0.4 | 103.6± 23.0 | 111.0± 0.2 | 111.4± 0.3 | 112.1± 0.4 |
| Med.-E | HC | 89.4±7.2 | 38.4±8.4 | **92.2±0.9** | 92.7± 0.8 | 91.1± 3.4 | 91.4± 2.5 |
| | HP | 95.5±9.4 | 88.4±15.9 | 14.6± 3.4 | **98.6± 10.7** | 78.4± 19.0 | 94.5±14.9 |
| | WK | 110.2±0.3 | 109.2±1.9 | **111.9± 0.6** | 109.2± 0.3 | 109.9± 0.4 | 110.1± 0.7 |
| Med.-R | HC | 44.7±0.4 | 46.9±0.3 | 41.3± 0.8 | 40.9± 0.6 | 45.2± 0.8 | **46.7± 0.6** |
| | HP | 73.8±18.9 | 95.5±1.7 | 86.4± 18.9 | 38.2± 9.4 | 89.2± 8.1 | **98.5± 2.0** |
| | WK | 64.5±17.0 | 77.5±3.1 | 65.1± 18.5 | 65.0± 15.7 | 82.1± 3.8 | **86.7± 3.2** |
| Med. | HC | **48.2±0.3** | **48.2±0.4** | 42.7± 1.8 | 44.3± 0.2 | 47.5± 0.4 | **48.6± 0.4** |
| | HP | 61.0±4.2 | 77.4±4.0 | 79.1± 8.7 | 57.5± 3.0 | 71.2± 6.5 | **86.4± 7.6** |
| | WK | **84.7±1.3** | 81.5±2.5 | 83.2± 0.8 | 81.0± 2.5 | 79.4± 4.7 | 85.0± 0.8 |
| Rand. | HC | 11.5±0.6 | 24.1±1.5 | 9.1± 0.6 | 3.2± 1.3 | 15.8± 1.6 | 16.3± 1.2 |
| | HP | 8.7±0.3 | 2.2±1.9 | 7.6± 0.8 | 7.3± 0.9 | 12.0± 10.0 | **20.4± 9.8** |
| | WK | 1.4±1.9 | 4.3±7.9 | **6.0± 1.3** | 3.1± 1.0 | 2.5± 2.6 | 2.3± 2.0 |
| | Total | 1010.0 | 930.3 | 908.5 | 956.5 | 1026.5 | **1102.1** |
| Antmaze | umaze | **92.8± 2.7** | 73.0±4.9 | 0.0±0.0 | 49.8± 6.2 | 83.6 ±4.5 | **88.4 ± 3.6** |
| | umaze-d. | 45.0± 22.2 | 43.8±4.4 | 9.6± 12.2 | **53.8± 13.0** | 43.2± 7.8 | **52.8±7.9** |
| | med.-p. | 0.0± 0.0 | 9.0±6.4 | 0.0± 0.0 | 0.0± 0.0 | **77.0± 5.1** | 48.6±25.3 |
| | med.-d. | 0.0± 0.0 | 3.8±4.2 | 0.0± 0.0 | 0.0± 0.0 | **56.8± 27.2** | 59.2±29.4 |
| | large-p. | 0.0± 0.0 | 0.0±0.0 | 0.0± 0.0 | 0.0± 0.0 | 1.0± 1.3 | **5.2 ± 8.4** |
| | large-d. | 0.0± 0.0 | 0.0±0.4 | 0.0± 0.0 | 0.0± 0.0 | 4.8± 9.6 | **6.6 ± 5.2** |
| | Total | 137.8 | 129.6 | 9.6 | 103.6 | **266.4** | 260.8 |

stable manner as shown in Figure 2(b). Similarly, as shown in Figure 2(c), the policy learned by AWAC often slows down around the point $(0, 0)$ and fails to reach the goal. This behavior implies that AWAC attempted to average over multiple modes of the distribution. By contrast, the policy learned by V2AE successfully reaches one of the goals. As the main difference between AWAC and V2AE is the policy architecture, this result shows that the unimodal policy distribution fails to deal with the mutimodal data distribution, while a mixture policy employed in V2AE successfully dealt with it. The activation of the sub-policies is visualized in Figure 2(e). The color indicates the value of the discrete latent variable given by the gating policy $z^* = \arg\max_z Q_{\boldsymbol{w}}(\boldsymbol{s}, \boldsymbol{\mu}(\boldsymbol{s}, \boldsymbol{z}))$. Figure 2(d) shows that different sub-policies are activated for different regions, which indicates that V2AE appropriately divided the state-action space.

## 7.2 D4RL BENCHMARK TASKS

We evaluated the performance of the proposed method on the benchmarking tasks in D4RL. As baseline methods, we used TD3+BC, CQL (Kumar et al., 2020), AWAC, and easyBCQ (Brandfonbrener et al., 2021). EasyBCQ is a one-step RL version of Batch Constraint Q-learning (Fujimoto et al., 2019). All the baseline methods use double-clipped Q-learning for the critic in this experiment. For easyBCQ, the state normalization and double-clipped Q-learning are used in our implementation. The implementation of AWAC and V2AE was identical to those in the previous experiment, and the difference between AWAC and V2AE indicates the effect of the policy representation. In our evaluation, $|Z| = 8$ was used for V2AE and infoV2AE. The effect of the dimensionality of the discrete latent variable is shown in Appendix D. In this study, we evaluated the baseline methods with antmaze-v0, Kitchen, Adroit, and mujoco-v2 tasks. For completeness, we provide the results with the mujoco-v0 datasets in Appendix E.

**Performance scores** Comparisons with the baseline methods are shown in Tables 3 and 4. The bold text indicates the best performance; the underlined text indicates the tasks for which V2AE outperformed AWAC. In mujoco-v2 tasks, we can see that the performance of V2AE is comparable/superior to the state-of-the-art methods on mujoco-v2 tasks. Regarding the comparison between V2AE and AWAC, the performance of V2AE matched or exceeded that of AWAC except the Hopper-expert and Hopper-medium-expert tasks. This result also confirms that the use of a mixture policy is beneficial for these tasks, although the benefits of using a mixture policy should

Table 4: Results on Kitchen and Adroit tasks using the average normalized scores over the last 10 test episodes and five seeds.

| | | TD3+BC (re-run) | CQL($\rho$) (paper) | easyBCQ (re-run) | AWAC (ours) | V2AE (ours) | infoV2AE (ours) |
|---|---|---|---|---|---|---|---|
| **Kitchen** | complete | $3.5 \pm 7.0$ | 31.3 | $17.0 \pm 8.9$ | $48.0 \pm 14.4$ | $\underline{\mathbf{59.0 \pm 12.5}}$ | $49.5 \pm 16.2$ |
| | mixed | $0.0 \pm 0.0$ | 52.4 | $10.0 \pm 5.7$ | $\mathbf{55.0 \pm 6.9}$ | $51.0 \pm 9.8$ | $39.5 \pm 10.3$ |
| | partial | $0.0 \pm 0.0$ | $\mathbf{50.1}$ | $12.0 \pm 11.4$ | $38.0 \pm 8.3$ | $38.5 \pm 6.0$ | $31.0 \pm 4.6$ |
| | Total | $3.5 \pm 7.0$ | 133.8 | $39.0 \pm 26.0$ | $141.0 \pm 29.6$ | $\mathbf{148.5 \pm 28.3}$ | $120 \pm 31.1$ |
| **Human** | pen | $0.8 \pm 8.0$ | 55.8 | $68.7 \pm 8.4$ | $65.3 \pm 20.4$ | $\underline{86.1 \pm 8.8}$ | $\mathbf{94.8 \pm 16.5}$ |
| | hammer | $0.9 \pm 0.8$ | 2.1 | $\mathbf{2.3 \pm 2.4}$ | $1.7 \pm 0.5$ | $\underline{1.2 \pm 0.2}$ | $\mathbf{2.4 \pm 0.9}$ |
| | door | $-0.3 \pm 0.0$ | $\mathbf{9.1}$ | $3.4 \pm 4.1$ | $6.0 \pm 4.2$ | $1.3 \pm 1.5$ | $4.2 \pm 3.1$ |
| | relocate | $-0.3 \pm 0.0$ | 0.0 | $\mathbf{0.2 \pm 0.3}$ | $0.0 \pm 0.0$ | $0.0 \pm 0.1$ | $\mathbf{0.1 \pm 0.0}$ |
| | Total | $1.1 \pm 8.8$ | 67.0 | $74.6 \pm 15.2$ | $73.0 \pm 25.1$ | $\underline{88.6 \pm 10.6}$ | $\mathbf{97.3 \pm 20.5}$ |
| **Cloned** | pen | $0.5 \pm 7.0$ | 40.3 | $\mathbf{59.9 \pm 29.7}$ | $17.7 \pm 14.9$ | $\underline{36.0 \pm 17.7}$ | $46.4 \pm 16.7$ |
| | hammer | $0.2 \pm 0.0$ | $\mathbf{5.7}$ | $0.4 \pm 0.0$ | $0.4 \pm 0.0$ | $\underline{0.8 \pm 0.6}$ | $1.2 \pm 0.3$ |
| | door | $-0.3 \pm 0.0$ | 3.5 | $0.0 \pm 0.0$ | $1.0 \pm 1.2$ | $0.0 \pm 0.0$ | $0.8 \pm 0.8$ |
| | relocate | $-0.3 \pm 0.0$ | -0.1 | $-0.3 \pm 0.0$ | $-0.2 \pm 0.0$ | $-0.2 \pm 0.0$ | $-0.2 \pm 0.0$ |
| | Total | $0.1 \pm 7.0$ | 42.4 | $\mathbf{60.0 \pm 29.7}$ | $18.9 \pm 16.1$ | $36.1 \pm 18.3$ | $48.2 \pm 15.8$ |

be task-dependent. Remarkably, our implementation of AWAC showed significantly better performance than that reported by Nair et al. (2020) because we employed the double-clipped Q-learning and state normalization. We provide the comparison between the original results of AWAC and ours in Appendix F. In addition, infoV2AE, which employs the MI-based regularization, outperformed V2AE on various tasks, and infoV2AE showed the best performance in terms of the sum of the overall scores on mujoco-v2. This result shows that encouraging the diversity of the sub-policies using the proposed MI-based regularization is effective for V2AE. The advantage of V2AE and infoV2AE over baseline methods is apparent for Antmaze tasks, which involve dealing with long horizons and require "stitching" together sub-trajectories in a given dataset (Fu et al., 2020). TD3+BC, CQL, easyBCQ, and AWAC did not work well on Antmaze tasks, and this result indicates that techniques used in these algorithm are not effective for such tasks. As the difference between V2AE and AWAC indicates the effect of different policy representations, the results indicate that the use of the mixture policy improves the performance to deal with tasks that require stitching together sub-trajectories in a given dataset. A comparison with additional baseline methods is provided in Appendix G. The effect of the scaling parameter $\alpha$ in V2AE and infoV2AE is reported in Appendix H.

**Critic loss function** To investigate the effect of the policy structure on the critic loss function, we compare the value of the critic loss function between AWAC and V2AE. In addition, to see the differences between the discrete and continuous latent variables, we also evaluated a variant of V2AE with the continuous latent variables, which we refer to as cV2AE. The normalized scores and the value of the critic loss function during training are shown in Figure 3. The value of the critic loss given by equation 13 is plotted for every 5,000 updates. The shaded area that indicates the error bar of the critic loss is removed for cV2AE on walker2d-medium-expert-v2 because the critic loss of cV2AE exploded. Previous studies indicated that the value of the critic loss can accumulate over iterations (Brandfonbrener et al., 2021). Figure 3 shows the accumulation of the critic loss in AWAC on the mujoco-v2 tasks. Importantly, in V2AE, the value of the critic loss is clearly lower, and the performance of the policy is better than that in AWAC. Brandfonbrener et al. (2021) showed that the accumulation of the critic loss can be reduced by introducing regularization. Since the difference between AWAC and V2AE is the policy representation, our results indicate that the use of the mixture policy can also mitigate the accumulation of the critic loss in offline RL. This result suggests the importance of incorporating inductive bias in the policy structure. However, it is worth noting that the reduction of the critic loss given by equation 13 does not necessarily lead to the improved performance of the policy. In halfcheetah-medium-expert-v2, although the critic loss was significantly lower in V2AE than that in AWAC, there was no significant difference in performance between V2AE and AWAC. Recently, Fujimoto et al. (2022) indicated that a lower value of the critic loss given by equation 13 does not necessarily lead to better performance, and the observation in Fujimoto et al. (2022) fits with what we observed in our experiments. Regarding the use of the continuous latent variable, while cV2AE often achieves lower values of the critic loss than AWAC, the critic loss of cV2AE occasionally explodes. A possible explanation for these results may be that learning the continuous latent variable provides flexibility to a policy representation but separate

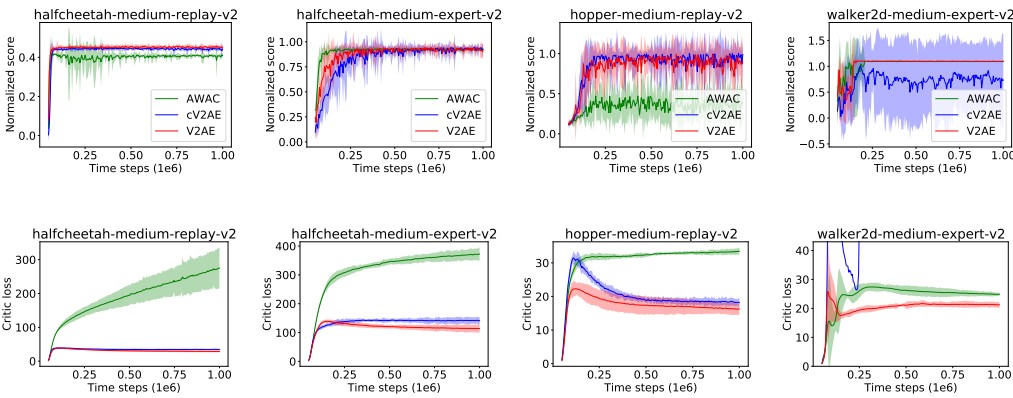

Figure 3: Critic loss and normalized score during the training.

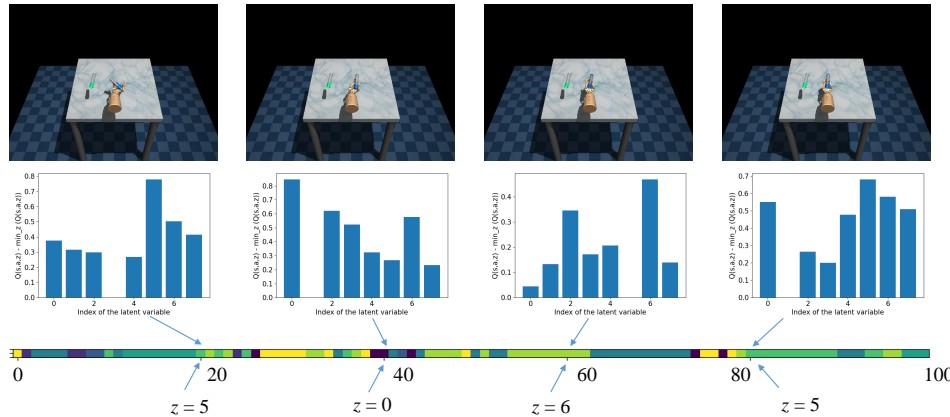

Figure 4: Visualization of the activation of sub-policies in the pen-human-v0 task. The top row shows the state at the 20th, 40th, 60th, and 80th time steps; the graphs in the middle row show the action-values of each of the sub-policies at each state.

modes of the objective function are still interpolated in the learned latent space, which results in generating out-of-distribution samples. The performance of cV2AE on mujoco-v2 tasks is reported in Appendix I, and more detailed results on the critic loss are provided in Appendix J.

**Qualitative evaluation of the learned latent variable** The activation of sub-policies in V2AE on the pen-human-v0 task is shown in Figure 4. In Figure 4, the top row shows the state at the 20th, 40th, 60th, and 80th time steps; the graphs in the middle row of the figure show the action-values of each of the sub-policies at each state, $Q_{\boldsymbol{w}}(\boldsymbol{s}, \boldsymbol{\mu}(\boldsymbol{s}, \boldsymbol{z}))$. A previous study on the option framework (Smith et al., 2018) reported that in the existing method only a few options are activated and that the remainder of the options do not learn meaningful behaviors. In contrast, the results in Figure 4 show that the value of each of the sub-policies $Q_{\boldsymbol{w}}(\boldsymbol{s}, \boldsymbol{\mu}(\boldsymbol{s}, \boldsymbol{z}))$ changes over time, and various sub-policies are activated during the execution. A figure showing the activation of sub-policies in different episodes is provided in Appendix K.

## 8 CONCLUSION

We have proposed an algorithm, which we call Value-Weighted Variational Auto-Encoder (V2AE), for training a mixture policy in offline RL. The V2AE algorithm can be interpreted as an approach that divides the state-action space by learning the discrete latent variable and learns the corresponding sub-policies in each region. Our experimental results show that the use of the mixture policy can mitigate the issue of critic error accumulation in offline RL. In addition, the experimental results also indicate that the use of the mixture policy significantly improves the performance of an offline RL algorithm. We believe that our work represents an important step toward leveraging the policy structure in offline RL.

REPRODUCIBILITY STATEMENT

To ensure the reproducibility of the results, we will provide the codes by posting a comment directed to the reviewers and area chairs and putting a link to an anonymous repository after the discussion forums are opened. We also summarized how to implement the proposed method and its variants in Section 6. More detailed descriptions of the implementation can be found in Appendix L. We used open-sourced benchmark tasks in D4RL to ensure reproducibility.

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

# A APPLICABILITY OF THE GATING POLICY

In the proposed algorithm, we employ the gating policy that determines the value of the latent variable as follows:

$$z = \arg\max_{z'} Q_w(s, \mu_\theta(s, z')), \tag{16}$$

where $\mu_\theta(s, z')$ represents the deterministic sub-policy, and $Q_w(s, a)$ is the approximated Q-function. While this gating policy looks specific to the case where $Q^\pi(s, a)$ is maximized, this gating policy is applicable to other objective function such as $A^\pi(s, a)$, $\exp(Q^\pi(s, a))$, and $\exp(A^\pi(s, a))$. The advantage function is defined as $A^\pi(s, a) = Q^\pi(s, a) - V^\pi(s)$. As the state value function $V^\pi(s)$ is independent from the action, we can obtain the following equation:

$$\arg\max_a Q^\pi(s, a) = \arg\max_a (Q^\pi(s, a) - V^\pi(s)) \tag{17}$$

$$= \arg\max_a A^\pi(s, a). \tag{18}$$

Thus, we can rewrite the gating policy as

$$z = \arg\max_{z'} Q_w(s, \mu_\theta(s, z')) \tag{19}$$

$$= \arg\max_{z'} A_w(s, \mu_\theta(s, z')). \tag{20}$$

Similarly, the exponential function $\exp(\cdot)$ is the monotonically increasing function. Thus, the extrema of $Q^\pi(s, a)$ is also the extrema of $\exp(Q^\pi(s, a))$. Thus, we can also rewrite the gating policy as

$$z = \arg\max_{z'} Q_w(s, \mu_\theta(s, z')) \tag{21}$$

$$= \arg\max_{z'} \exp\left(Q_w(s, \mu_\theta(s, z'))\right) \tag{22}$$

$$= \arg\max_{z'} A_w(s, \mu_\theta(s, z')) \tag{23}$$

$$= \arg\max_{z'} \exp\left(A_w(s, \mu_\theta(s, z'))\right). \tag{24}$$

As the latent variable is discrete, we can analytically compute $\arg\max_{z'} Q_w(s, \mu_\theta(s, z'))$. As we used this gating policy, the gating policy is deterministic in our implementation.

# B DERIVATION OF THE VARIATIONAL LOWER BOUND

We employed the variational lower bound in equation 7 to derive the objective function for the proposed method. We provide the detailed derivation, which was omitted in the main manuscript due to the page limitation. We denote by $p(\cdot)$ the true distribution induced by the policy $\pi_\theta(a|s)$, and the distribution that approximates the true distribution is denoted by $q(\cdot)$. The KL divergence between $q(x)$ and $p(x)$ is defined as

$$D_{\mathrm{KL}}\big(q(x)||p(x)\big) = \int q(x) \log \frac{q(x)}{p(x)} dz. \tag{25}$$

Based on the above notation, the log-likelihood $\log \pi_\theta(a_i|s_i)$ can be written as follows:

$$\log \pi_\theta(a_i|s_i) = \int q_\phi(z|s_i, a_i) \log \pi_\theta(a_i|s_i) dz \tag{26}$$

$$= \int q_\phi(z|s_i, a_i)\big(\log \pi(a_i|s_i, z) + \log p(z|s_i) - \log p(z|s_i, a_i)\big) dz \tag{27}$$

$$= \int q_\phi(z|s_i, a_i) \log \frac{q(z|s_i, a_i)}{p(z|s_i, a_i)} dz - \int q(z|s_i, a_i) \log \frac{q(z|s_i, a_i)}{p(z|s_i)} dz \tag{28}$$

$$+ \int q(z|s_i, a_i) \log \pi_\theta(a_i|s_i, z) dz \tag{29}$$

$$= D_{\mathrm{KL}}\big(q(z|s_i, a_i)||p(z|s_i, a_i)\big) - D_{\mathrm{KL}}(q(z|s_i, a_i)||p(z|s_i))$$
$$+ \mathbb{E}_{z \sim q(z|s_i, a_i))}\left[\log \pi_\theta(a_i|s_i, z)\right]. \tag{30}$$

Table 5: Effect of the dimensionality of the discrete latent variable on Adroit tasks.

| | infoV2AE $|Z| = 4$ | infoV2AE $|Z| = 8$ | infoV2AE $|Z| = 16$ | infoV2AE $|Z| = 32$ |
|---|---|---|---|---|
| pen-human-v0 | $75.7 \pm 18.9$ | $\mathbf{94.8 \pm 16.5}$ | $75.0 \pm 17.5$ | $86.7 \pm 12.4$ |
| kitchen-complete-v0 | $30.0 \pm 13.5$ | $35.5 \pm 29.7$ | $\mathbf{49.5 \pm 16.2}$ | $42.5 \pm 20.8$ |
| walker2d-expert-v2 | $99.7\pm17.9$ | $\mathbf{112.1\pm0.4}$ | $108.8\pm6.8$ | $\mathbf{106.4\pm10.2}$ |
| walker2d-medium-expert-v2 | $89.1\pm25.7$ | $\mathbf{110.1\pm0.7}$ | $96.0\pm17.0$ | $\mathbf{109.9\pm0.6}$ |
| walker2d-medium-replay-v2 | $81.6\pm4.5$ | $\mathbf{86.7\pm3.2}$ | $85.4\pm3.7$ | $\mathbf{86.3\pm3.1}$ |
| walker2d-medium-v2 | $81.8\pm2.5$ | $\mathbf{85.0\pm0.8}$ | $69.9\pm28.3$ | $\mathbf{84.3\pm1.0}$ |

In the first line, we consider the marginalization over $\boldsymbol{z}$. As $\log \pi(\boldsymbol{a}|\boldsymbol{s})$ is independent from the latent variable $\boldsymbol{z}$, the equality in the first line holds. As $D_{\mathrm{KL}}\big(q(\boldsymbol{z}|\boldsymbol{s},\boldsymbol{a})||p(\boldsymbol{z}|\boldsymbol{s},\boldsymbol{a})\big) > 0$, we can obtain a variant of the variational lower bound of the conditional log-likelihood:

$$\log \pi_{\boldsymbol{\theta}}(\boldsymbol{a}_i|\boldsymbol{s}_i) \geq -D_{\mathrm{KL}}(q_{\boldsymbol{\phi}}(\boldsymbol{z}|\boldsymbol{s}_i,\boldsymbol{a}_i)||p(\boldsymbol{z}|\boldsymbol{s}_i)) + \mathbb{E}_{\boldsymbol{z}\sim q(\boldsymbol{z}|\boldsymbol{s}_i,\boldsymbol{a}_i))}\left[\log \pi_{\boldsymbol{\theta}}(\boldsymbol{a}_i|\boldsymbol{s}_i,\boldsymbol{z})\right]. \tag{31}$$

## C PROOF OF THE CONTRACTION OF THE LATENT-MAX-Q OPERATOR

We consider the operator $\mathcal{T}_{\boldsymbol{z}}$, which is given by

$$\mathcal{T}_{\boldsymbol{z}}Q(\boldsymbol{s},\boldsymbol{a}) = \mathbb{E}_{\boldsymbol{s}'}\left[r(\boldsymbol{s},\boldsymbol{a}) + \gamma \max_{\boldsymbol{z}} Q(\boldsymbol{s}',\boldsymbol{\mu}(\boldsymbol{s}',\boldsymbol{z}'))\right]. \tag{32}$$

To prove the contraction of $\mathcal{T}_{\boldsymbol{z}}$, we use the infinity norm given by

$$\|Q_1 - Q_2\|_\infty = \max_{\boldsymbol{s}\in\mathcal{S},\boldsymbol{a}\in\mathcal{A}} |Q_1(\boldsymbol{s},\boldsymbol{a}) - Q_2(\boldsymbol{s},\boldsymbol{a})|, \tag{33}$$

where $Q_1$ and $Q_2$ are different estimates of the Q-function. We consider the infinity norm of the difference between these two estimates, $Q_1$ and $Q_2$, after applying the operator $\mathcal{T}_{\boldsymbol{z}}$:

$$\|\mathcal{T}_{\boldsymbol{z}}Q_1 - \mathcal{T}_{\boldsymbol{z}}Q_2\|_\infty \tag{34}$$

$$= \left|\mathbb{E}_{\boldsymbol{s}'}\left[r(\boldsymbol{s},\boldsymbol{a}) + \gamma \max_{\boldsymbol{z}'} Q_1(\boldsymbol{s}',\boldsymbol{\mu}(\boldsymbol{s}',\boldsymbol{z}'))\right] - \mathbb{E}_{\boldsymbol{s}'}\left[r(\boldsymbol{s},\boldsymbol{a}) + \gamma \max_{\boldsymbol{z}'} Q_2(\boldsymbol{s}',\boldsymbol{\mu}(\boldsymbol{s}',\boldsymbol{z}'))\right]\right| \tag{35}$$

$$= \left|\gamma\mathbb{E}_{\boldsymbol{s}'}\left[\max_{\boldsymbol{z}'} Q_1(\boldsymbol{s}',\boldsymbol{\mu}(\boldsymbol{s}',\boldsymbol{z}'))\right] - \gamma\mathbb{E}_{\boldsymbol{s}'}\left[\max_{\boldsymbol{z}'} Q_2(\boldsymbol{s}',\boldsymbol{\mu}(\boldsymbol{s}',\boldsymbol{z}'))\right]\right| \tag{36}$$

$$= \gamma\left|\mathbb{E}_{\boldsymbol{s}'}\left[\max_{\boldsymbol{z}'} Q_1(\boldsymbol{s}',\boldsymbol{\mu}(\boldsymbol{s}',\boldsymbol{z}'))\right] - \mathbb{E}_{\boldsymbol{s}'}\left[\max_{\boldsymbol{z}'} Q_2(\boldsymbol{s}',\boldsymbol{\mu}(\boldsymbol{s}',\boldsymbol{z}'))\right]\right| \tag{37}$$

$$= \gamma\left|\mathbb{E}_{\boldsymbol{s}'}\left[\max_{\boldsymbol{z}'} Q_1(\boldsymbol{s}',\boldsymbol{\mu}(\boldsymbol{s}',\boldsymbol{z}')) - \max_{\boldsymbol{z}'} Q_2(\boldsymbol{s}',\boldsymbol{\mu}(\boldsymbol{s}',\boldsymbol{z}'))\right]\right| \tag{38}$$

$$\leq \gamma\left|\mathbb{E}_{\boldsymbol{s}'}\|Q_1 - Q_2\|_\infty\right| \tag{39}$$

$$\leq \gamma\|Q_1 - Q_2\|_\infty. \tag{40}$$

The above relationship shows the contraction of the operator $\mathcal{T}_{\boldsymbol{z}}$.

## D EFFECT OF THE DIMENSIONALITY OF THE DISCRETE LATENT VARIABLE

In our evaluation, we first evaluated the effect of the dimensionality of the discrete latent variable. The results are shown in Table 5. As $|Z| = 8$ consistently showed satisfactory performance, $|Z| = 8$ was used in the subsequent evaluations.

As shown, infoV2AE with $|Z| = 8$ demonstrated the best performance, while the performance with $|Z| = 16$ and $|Z| = 32$ is comparable. These results show that the performance of the policy is not so sensitive to the dimensionality of the latent varialbe. However, the performance with $|Z| = 4$ is relatively weak, and it indicates that the policy may not be expressive enough when the dimensionality of the latent varialbe is too small.

Table 6: Results on Mujoco tasks using D4RL-v0 datasets. The average normalized scores after 1 million time steps over the last 10 test episodes and five seeds are shown. HC = HalfCheetah, HP = Hopper, WK = Walker2d, Med.-E = Mediaum-Expert, Med.-R = Medium-Replay, Med.=Medium, and Rand.=Random. The bold text indicates the best performance; the underlined text indicates the tasks for which V2AE outperformed AWAC. $|Z| = 8$ for V2AE.

|  |  | TD3+BC (re-run) | CQL (re-run) | easyBCQ (ours) | AWAC (ours) | V2AE (ours) |
|---|---|---|---|---|---|---|
| Expert | HC | $104.9 \pm 2.9$ | $95.9\pm8.9$ | $101.8\pm 0.8$ | $\mathbf{107.3\pm 0.3}$ | $103.9\pm 1.9$ |
|  | HP | $\mathbf{112.1 \pm 0.3}$ | $\mathbf{112.4\pm0.4}$ | $108.4 \pm 1.8$ | $\mathbf{112.0\pm 0.2}$ | $\mathbf{112.2\pm 0.2}$ |
|  | WK | $\mathbf{105.8 \pm 2.1}$ | $87.8 \pm 31.1$ | $83.1 \pm 19.3$ | $104.4\pm 3.2$ | $\mathbf{105.5\pm 3.7}$ |
| Med.-E | HC | $\mathbf{102.3 \pm 3.8}$ | $23.8 \pm 12.4$ | $41.2\pm 1.8$ | $\mathbf{101.7\pm 4.9}$ | $83.9\pm 20.4$ |
|  | HP | $111.9 \pm 0.1$ | $\mathbf{112.3 \pm 1.0}$ | $103.7 \pm 13.5$ | $111.9\pm 0.1$ | $\mathbf{112.2\pm 0.1}$ |
|  | WK | $96.7 \pm 21.7$ | $\mathbf{100.6 \pm 11.2}$ | $93.4 \pm 10.2$ | $101.7\pm 7.3$ | $\underline{103.3\pm 2.5}$ |
| Med.-R | HC | $43.4 \pm 0.3$ | $45.3 \pm 0.4$ | $45.2\pm 1.4$ | $40.7\pm 0.3$ | $\underline{42.6\pm 0.4}$ |
|  | HP | $\mathbf{33.5 \pm 3.9}$ | $29.8 \pm 2.8$ | $\mathbf{32.2 \pm 2.0}$ | $27.7\pm 1.3$ | $\underline{\mathbf{32.0\pm 4.8}}$ |
|  | WK | $14.1 \pm 4.0$ | $16.7 \pm 4.0$ | $16.8 \pm 1.6$ | $13.9\pm 2.0$ | $\underline{17.3\pm 2.4}$ |
| Med. | HC | $43.7 \pm 0.4$ | $40.7\pm0.6$ | $43.5\pm 0.6$ | $40.0\pm 0.4$ | $\underline{41.9\pm 0.3}$ |
|  | HP | $\mathbf{99.8 \pm 0.3}$ | $47.0\pm26.0$ | $37.4 \pm 10.0$ | $30.2\pm 0.3$ | $\underline{57.9\pm 25.6}$ |
|  | WK | $81.4 \pm 1.0$ | $69.3\pm24.1$ | $78.5 \pm 0.7$ | $53.8\pm 9.9$ | $\underline{62.0\pm 13.4}$ |
| Rand. | HC | $13.0 \pm 0.4$ | $\mathbf{28.5\pm 1.7}$ | $8.4\pm 0.5$ | $2.5\pm 0.3$ | $\underline{17.1\pm 1.2}$ |
|  | HP | $11.0 \pm 0.1$ | $10.7\pm0.1$ | $\mathbf{11.6 \pm 0.2}$ | $10.6\pm 0.1$ | $\underline{\mathbf{11.5\pm 0.5}}$ |
|  | WK | $1.3 \pm 1.2$ | $1.5\pm3.2$ | $5.2 \pm 0.2$ | $3.2\pm 1.7$ | $2.0\pm 2.5$ |

# E    PERFORMANCE ON D4RL-V0 DATASETS

As reported in (Fujimoto & Gu, 2021), there is a non-negligible difference when using the D4RL-v2 and D4RL-v0 datasets. Having presented our results using D4RL-v2 datasets in the main manuscript, we here show the results using D4RL-v0 datasets in Table 6. For CQL, we show the results obtained by re-running the codes given in the website[1]. We performed the experiments with our implementations of easyBCQ and AWAC. The results were mixed, and it is difficult to identify the best algorithm for the MuJoCo tasks with the D4RL-v0 datasets. However, we can see that the performance of V2AE was comparable to the state-of-the-art methods on the MuJoCo tasks with the D4RL-v0 datasets. Especially, compared to the baseline methods, the performance of V2AE consistently matched or exceeded the performance of the other methods on datasets containing expert demonstrations, e.g., *-expert and *-medium-expert.

# F    PERFORMANCE OF AWAC

We used two techniques in our implementation of AWAC, which are not used in the original paper of AWAC (Nair et al., 2020): 1) state normalization proposed in (Fujimoto & Gu, 2021) and 2) normalization of the advantage function in equation 15. We provide the comparison between the performance of our implementation of AWAC and the performance reported in the original paper (Nair et al., 2020). Our experiments show that our implementation significantly outperformed the results reported in the original paper of AWAC.

# G    COMPARISON WITH ADDITIONAL BASELINES

We provide comparison with additional baselines on mujoco-v2 tasks in D4RL in Table 8. We show the results of MAPLE, which is a recent model-based offline algorithm using latent representations (Chen et al., 2021b). In addition, we show the results of Decision Transformer (Chen et al., 2021a), as a representative of transformer-based methods. We also included Implicit Q-learning (IQL), which employs expectile regression for learning the Q-function Kostrikov et al. (2022). While these methods are well-known and state-of-the-art, we omit them in the main manuscript due to the page limitation, and we focused on methods that use double-clipped Q-learning for the critic in the

---

[1]https://github.com/aviralkumar2907/CQL

Table 7: Comparison of AWAC implementations on Mujoco tasks using D4RL-v0 datasets. The average normalized scores after 1 million time steps over the last 10 test episodes and five seeds are shown. The bold text indicates the tasks for which the performance of our implemnetation of AWAC exceeded the that of AWAC reported in the original paper (Nair et al., 2020). HC = HalfCheetah, HP = Hopper, WK = Walker2d.

|  |  | AWAC (paper) | AWAC (ours) |
|---|---|---|---|
| Expert | HC | 78.5 | **107.3± 0.3** |
| | HP | 85.2 | **112.0± 0.2** |
| | WK | 57.0 | **104.4± 3.2** |
| Med.-E | HC | 36.8 | **101.7± 4.9** |
| | HP | 80.9 | **111.9± 0.1** |
| | WK | 42.7 | **101.7± 7.3** |
| Med.-R | HC | - | 40.7± 0.3 |
| | HP | - | 27.7± 1.3 |
| | WK | - | 13.9± 2.0 |
| Med. | HC | 37.4 | **40.0± 0.4** |
| | HP | 72.0 | 30.2± 0.3 |
| | WK | 30.1 | **53.8± 9.9** |
| Rand. | HC | 2.2 | 2.5± 0.3 |
| | HP | 9.6 | **10.6± 0.1** |
| | WK | 5.1 | 3.2± 1.7 |

Table 8: Results on MuJoCo tasks using D4RL-v2 datasets. Average normalized scores over the last 10 test episodes and five seeds are shown. HC = HalfCheetah, HP = Hopper, WK = Walker2d. The gray text indicates the performance lower than that of V2AE/infoV2AE. The bold text indicates the best performance.

|  |  | MAPLE (paper) | Decision Transformer (paper) | IQL (paper) | V2AE (ours) | infoV2AE (ours) |
|---|---|---|---|---|---|---|
| Med.-E | HC | 63.5 ± 6.5 | 86.8 ± 1.3 | 86.7 | **91.1± 3.4** | **91.4± 2.5** |
| | HP | 42.5 ± 4.1 | **107.6 ± 1.8** | 91.5 | 78.4± 19.0 | 94.5±14.9 |
| | WK | 73.8 ± 8.0 | 108.1 ± 0.2 | 109.6 | **109.9± 0.4** | **110.1± 0.7** |
| Med.-R | HC | **59.0 ± 0.6** | 36.6 ± 0.8 | 44.2 | 45.2± 0.8 | 46.7± 0.6 |
| | HP | 87.5 ± 10.8 | 82.7 ± 7.0 | 94.7 | 89.2± 8.1 | **98.5± 2.0** |
| | WK | 76.7 ± 3.8 | 66.6 ± 3.0 | 73.9 | 82.1± 3.8 | **86.7± 3.2** |
| Med. | HC | **50.4 ± 1.9** | 42.6 ± 0.1 | 47.4 | 47.5± 0.4 | 48.6± 0.4 |
| | HP | 21.1 ± 1.2 | 67.6 ± 1.0 | 66.3 | 71.2± 6.5 | **86.4± 7.6** |
| | WK | 56.3 ± 10.6 | 74.0 ± 1.4 | 78.3 | 79.4± 4.7 | **85.0± 0.8** |

main manuscript. For each baseline methods, we adapted the results reported in the original paper. V2AE and infoV2AE show performance consistently better than or comparable to these baseline methods, although our implementation of V2AE and infoV2AE do not employ techniques such as ensemble of critics. This result indicates the significant effect on the policy structure in offline RL.

### G.1 COMPARISON WITH AWAC USING A GAUSSIAN MIXTURE POLICY

We also provide the comparison with AWAC with a Gaussian mixture policy in Table 9. Overall, the use of the Gaussian mixture policy does not improve the performance of AWAC. A recent study by Chen et al. (2022) indicates that the use of the Gaussian mixture policy does not improve the performance of AWAC. When a Gaussian mixture policy is employed, a Gaussian component that covers a large part of the state space often appears, and it governs the resulting performance. This behavior is also often observed in the context of the option critic for online RL (see Smith et al. (2018)). If that happens, we cannot exploit the discrete latent variable.

By contrast, we employed a mixture of deterministic policies, not a mixture of the Gaussian policies. Unlike a Gaussian policy, a deterministic policy can be seen as a distribution given by the dirac-delta function, which is the limit of the Gaussian as the standard deviation goes zero. When we use the mixture of deterministic policies, we will not have a component that covers the large state space.

Table 9: Comparison with AWAC with a Gaussian mixture policy (mixAWAC) using D4RL-v2 datasets. Average normalized scores over the last 10 test episodes and five seeds are shown. The bold text indicates the best performance.

| | | mixAWAC | AWAC | V2AE | infoV2AE |
|---|---|---|---|---|---|
| **Expert** | HalfCheetah | 94.0±0.5 | 94.8± 0.2 | **97.0± 1.0** | **95.6± 2.0** |
| | Hopper | **111.8±0.8** | 109.8± 2.9 | 93.6± 15.1 | 107.5± 2.9 |
| | Walker2d | 110.5±0.3 | 111.0± 0.2 | 111.4± 0.3 | **112.1± 0.4** |
| **Med.-E** | HalfCheetah | 92.1±0.6 | **92.7± 0.8** | 91.1± 3.4 | 91.4± 2.5 |
| | Hopper | **102.0±17.5** | 98.6± 10.7 | 78.4± 19.0 | 94.5±14.9 |
| | Walker2d | 109.1±0.3 | 109.2± 0.3 | 109.9± 0.4 | **110.1± 0.7** |
| **Med.-R** | HalfCheetah | 41.5±0.4 | 40.9± 0.6 | 45.2± 0.8 | **46.7± 0.6** |
| | Hopper | 41.2±4.7 | 38.2± 9.4 | 89.2± 8.1 | **98.5± 2.0** |
| | Walker2d | 67.7±8.8 | 65.0± 15.7 | 82.1± 3.8 | **86.7± 3.2** |
| **Med.** | HalfCheetah | 45.1±0.3 | 44.3± 0.2 | 47.5± 0.4 | **48.6± 0.4** |
| | Hopper | 57.2±3.9 | 57.5± 3.0 | 71.2± 6.5 | **86.4± 7.6** |
| | Walker2d | 78.7 ± 4.8 | 81.0± 2.5 | 79.4± 4.7 | **85.0± 0.8** |
| **Rand.** | HalfCheetah | 2.2±0.0 | 3.2± 1.3 | 15.8± 1.6 | **16.3± 1.2** |
| | Hopper | 8.2±0.2 | 7.3± 0.9 | 12.0± 10.0 | **20.4± 9.8** |
| | Walker2d | **4.9±1.1** | 3.1± 1.0 | 2.5± 2.6 | 2.3± 2.0 |
| **Antmaze** | umaze-v0 | 57.4±6.2 | 49.8±6.2 | 83.6±4.5 | **88.4±3.6** |
| | umaze-diverse-v0 | 46.8±6.9 | **53.8±13.0** | 43.2± 7.8 | 52.8±7.9 |
| | medium-play-v0 | 0.0± 0.0 | 0.0±0.0 | **77.0±5.1** | 48.6±25.3 |
| | medium-diverse-v0 | 0.0± 0.0 | 0.0±0.0 | 56.8±27.2 | **59.2±29.4** |
| | large-play-v0 | 0.0± 0.0 | 0.0± 0.0 | 1.0±1.3 | **5.2± 8.4** |
| | large-diverse-v0 | 0.0± 0.0 | 0.0± 0.0 | 4.8±9.6 | **6.6± 5.2** |

The use of the mixture of deterministic policies allows us to distribute each component separately, and we can exploit the benefit of the discrete latent variable. In addition, we also leverage the variational lower bound to incorporate the prior distribution of the latent variable, which enables us to obtain meaningful latent representations. These are the reasons why our methods clearly outperformed AWAC, while simply using the Gaussian mixture policy does not improve the performance of AWAC.

## G.2 COMPARISON WITH LAPO

Recently, Chen et al. (2022) proposed an algorithm, Latent-variable Advantage-weighted Policy Optimization (LAPO), which leverage the continuous latent space for policy learning. As the approach is related to our method V2AE, we provide the detailed comparison in this section. We found that the authors' implementation of LAPO in `https://github.com/pcchenxi/LAPO-offlienRL` includes techniques to improve the performance, such as action normalization and clipping of the target value for the state-value function. Such techniques are compatible with V2AE and our baseline methods, but they are not incorporated in our experiments. For this reason, we first evaluated LAPO without these techniques, which we refer to as LAPO-.

The results are reported in Table 10. Overall, V2AE and infoV2AE outperformed LAPO on mujoco-v2 tasks and Antmaze-v0 tasks. While our method V2AE leverages the discrete latent space, LAPO uses the continuous latent space. In this sense, LAPO is closer to cV2AE, which is a variant of V2AE that uses the continuous latent space. In the main manuscript, we showed that the critic loss value often explodes during the training of cV2AE. Similarly, the critic log value increases rapidly at the beginning of the training in LAPO, as shown in Figure 5. While the critic log value often decreases at the end of the training of LAPO, the critic loss value is still higher than that of V2AE. As the performance of the policy is better in V2AE in these tasks, the surge of the critic loss value indicates the generation of the out-of-distribution actions during the training in LAPO. These results support our observations in the main manuscript and indicate that the use of the discrete latent variable is effective to reduce the generation of the out-of-distribution actions.

## G.3 COMPARISON WITH IQL AND LAPO ON ANTMAZE

As the Antmaze tasks are considered as challenging tasks on offline RL, we provide detailed results on the Antmaze tasks in this section. As we found that a few techniques used in LAPO are essential

Table 10: Comparison with AWAC with a Gaussian mixture policy (mixAWAC) using D4RL-v2 datasets. Average normalized scores over the last 10 test episodes and five seeds are shown. The bold text indicates the best performance.

| | | LAPO- | V2AE | infoV2AE |
|---|---|---|---|---|
| **Expert** | HalfCheetah | 94.9±0.2 | **97.0± 1.0** | 95.6± 2.0 |
| | Hopper | **111.4±0.6** | 93.6± 15.1 | 107.5± 2.9 |
| | Walker2d | 111.1±0.2 | 111.4± 0.3 | **112.1± 0.4** |
| **Med.-E** | HalfCheetah | 94.2±0.4 | 91.1± 3.4 | 91.4± 2.5 |
| | Hopper | **110.9±1.0** | 78.4± 19.0 | 94.5±14.9 |
| | Walker2d | 110.4±0.1 | 109.9± 0.4 | **110.1± 0.7** |
| **Med.-R** | HalfCheetah | 40.7±1.9 | 45.2± 0.8 | **46.7± 0.6** |
| | Hopper | 52.6±11.3 | 89.2± 8.1 | **98.5± 2.0** |
| | Walker2d | 60.1±11.7 | 82.1± 3.8 | **86.7± 3.2** |
| **Med.** | HalfCheetah | 45.5±0.2 | 47.5± 0.4 | **48.6± 0.4** |
| | Hopper | 51.8±3.6 | 71.2± 6.5 | **86.4± 7.6** |
| | Walker2d | 77.7±3.5 | 79.4± 4.7 | **85.0± 0.8** |
| **Rand.** | HalfCheetah | 2.3±0.3 | 15.8± 1.6 | **16.3± 1.2** |
| | Hopper | **21.8±11.3** | 12.0± 10.0 | 20.4± 9.8 |
| | Walker2d | 5.6±3.1 | 2.5± 2.6 | 2.3± 2.0 |
| **Antmaze** | umaze-v0 | 76.4±3.3 | 83.6±4.5 | **88.4±3.6** |
| | umaze-diverse-v0 | 56.6±8.6 | 43.2± 7.8 | **52.8±7.9** |
| | medium-play-v0 | 0.8±1.6 | **77.0±5.1** | 48.6±25.3 |
| | medium-diverse-v0 | 0.0±0.0 | **56.8±27.2** | 59.2±29.4 |
| | large-play-v0 | 0.0±0.0 | 1.0±1.3 | **5.2± 8.4** |
| | large-diverse-v0 | 0.0±0.0 | 4.8±9.6 | **6.6± 5.2** |

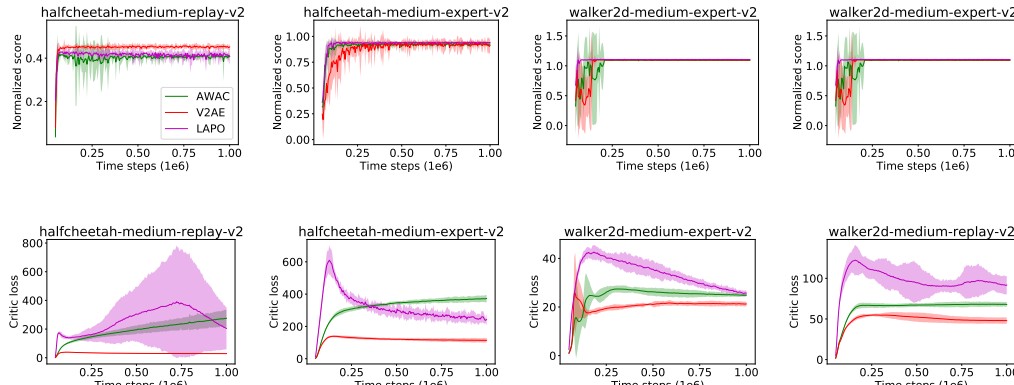

Figure 5: Critic loss and normalized score during training on mujoco-v2 locomotion tasks.

to reproduce the results in Chen et al. (2022), we also evaluated variants of V2AE, which incorporate such techniques. In methods named like "xx_lapo", the techniques used in LAPO Chen et al. (2022) are incorporated. We summarized the techniques/differences as below:

- computation of the target value as $Q_{\text{target}} = 0.7 \min(Q_1, Q_2) + 0.3 \max(Q_1, Q_2)$
- target value clipping $v_{\text{target}} = \min(\max(v_{\text{target}}, v_{\text{min}}), v_{\text{max}})$
- reward scaling $r \leftarrow r \times 100$
- network architecture: three hidden layers with 256 units, learning rate: 2e-4

For target value clipping, $v_{\text{min}}$ and $v_{\text{max}}$ are computed as $v_{\text{min}} = \min_{\mathcal{D}} r \cdot \frac{1}{1-\gamma}$ and $v_{\text{max}} = \max_{\mathcal{D}} r \cdot \frac{1}{1-\gamma}$, respectively. In LAPO-, IQL, V2AE, infoV2AE, we used the reward $r \leftarrow r - 1$, following the protocol used in the original IQL paper. While action normalization is used in LAPO, we found that the action normalization does not improve the performance of IQL and V2AE. Therefore, action normalization is used only in LAPO in our results. In the table below, LAPO- shows LAPO without these techniques, although the network architecture is the same as the original LAPO. "IQL (re-run)" indicates the results of re-running IQL in our code base, which fairly reproduced the results reported in the original IQL paper. "V2AE" shows the result reported in our main manuscript.

Table 11: Comparison with IQL on Antmaze tasks. Average normalized scores over the last 100 test episodes and five seeds are shown. The bold text indicates the best performance.

| | IQL (re-run) | IQL_lapo | V2AE | V2AE_lapo | infoV2AE | infoV2AE_lapo |
|---|---|---|---|---|---|---|
| umaze-v0 | 87.4±4.5 | 87.2±2.8 | 83.6 ±4.5 | 92.8±2.1 | 88.4±3.6 | 89.4±5.1 |
| umaze-diverse-v0 | 64.6±5.6 | 54.0±13.4 | 43.2± 7.8 | 32.6±25.8 | 52.8±7.9 | 34.8±18.0 |
| medium-play-v0 | 74.6±3.1 | 64.0±9.5 | 77.0± 5.1 | 63.0±13.0 | 48.6±25.3 | 62.6±6.8 |
| medium-diverse-v0 | 73.8±7.1 | 54.8±8.6 | 56.8± 27.2 | 75.0±8.5 | 59.2±29.4 | 82.8±4.4 |
| large-play-v0 | 39.0±7.2 | 20.4±11.0 | 1.0± 1.3 | 42.2±23.0 | 5.2±8.4 | 47.4±14.5 |
| large-diverse-v0 | 48.0±9.0 | 17.0±4.2 | 4.8± 9.6 | 56.6±4.5 | 6.6±5.2 | 38.0±4.8 |

Table 12: Comparison with LAPO on Antmaze tasks. Average normalized scores over the last 100 test episodes and five seeds are shown. The bold text indicates the best performance.

| | LAPO- | LAPO (re-run) | V2AE | V2AE_lapo | infoV2AE | infoV2AE_lapo |
|---|---|---|---|---|---|---|
| umaze-v0 | 76.4±3.3 | 97.8±1.3 | 83.6 ±4.5 | 92.8±2.1 | 88.4±3.6 | 89.4±5.1 |
| umaze-diverse-v0 | 56.6±8.6 | 52.4±20.5 | 43.2± 7.8 | 32.6±25.8 | 52.8±7.9 | 34.8±18.0 |
| medium-play-v0 | 0.8±1.6 | 64.2±5.3 | 77.0± 5.1 | 63.0±13.0 | 48.6±25.3 | 62.6±6.8 |
| medium-diverse-v0 | 0.0±0.0 | 69.4±12.8 | 56.8± 27.2 | 75.0±8.5 | 59.2±29.4 | 82.8±4.4 |
| large-play-v0 | 0.0±0.0 | 12.4±16.9 | 1.0± 1.3 | 42.2±23.0 | 5.2±8.4 | 47.4±14.5 |
| large-diverse-v0 | 0.0±0.0 | 18.2±16.1 | 4.8± 9.6 | 56.6±4.5 | 6.6±5.2 | 38.0±4.8 |

In Tables 11 and 12, we reported the results across five different seeds with 100 test episodes after 1 million updates. Regarding LAPO, we could not reproduce the results reported in the paper, which may be due to the difference of the experiment procedure. In the original paper (Chen et al., 2022), the methods are evaluated across three random seeds with 10 test episodes. When using the techniques used in LAPO, the overall performance of V2AE and infoV2AE was better than that of LAPO. While LAPO nicely showed how to leverage the latent space for offline RL, LAPO does not utilize the discrete latent space. The difference in performance between LAPO and our methods shows the benefit of leveraging the discrete latent variable.

Regarding IQL, we could fairly reproduce the performance reported in Kostrikov et al. (2022). Interestingly, the techniques used in LAPO did not improve the performance of IQL. When using the techniques used in LAPO, the overall performance of V2AE and infoV2AE was better or comparable to IQL. This result shows that incorporating the policy structure can achieve the performance achieved by using the state-of-the-art algorithm for the critic.

## H  EFFECT OF THE SCALING PARAMETER

To investigate the effect of the value of $\alpha$ in equation 15, we evaluated the performance of V2AE and infoV2AE with $\alpha = 5.0$ and $\alpha = 10.0$. The results are shown in Table 13. Except hopper-medium-expert-v2, results with $\alpha = 10.0$ are better than those with $\alpha = 5.0$. In the main manuscript, we report the result of $\alpha = 5.0$ for infoV2AE on hopper-medium-expert-v2, $\alpha = 10.0$ for the rest of the mujoco-v2 tasks.

## I  PERFORMANCE OF CV2AE

We provide the performance of cV2AE, which is a variant of V2AE with the continuous latent variable, on mujoco-v2 tasks in Table 14. cV2AE outperformed AWAC in terms of the sum of the score across the tasks. However, overall performance is lower than V2AE and infoV2AE. In cV2AE, a policy is more flexible and expressive than a Gaussian policy, which resulted in performance better than AWAC. Meanwhile, the continuous latent space still interpolates the separate modes of the multimodal distribution in cV2AE, and it will result in generating the out-of-distribution actions in offline RL. We think that is the reason why V2AE with the discrete latent variable outperforms cV2AE with the continuous latent variable.

Table 13: Results on mujoco tasks using D4RL-v2 datasets with different values of $\alpha$ in equation 15. Average normalized scores over the last 10 test episodes and five seeds are shown. HC = HalfCheetah, HP = Hopper, WK = Walker2d.

| | | V2AE $\alpha = 5.0$ | V2AE $\alpha = 10.0$ | infoV2AE $\alpha = 5.0$ | infoV2AE $\alpha = 10.0$ |
|---|---|---|---|---|---|
| Expert | HC | 95.9±1.0 | 97.0± 1.0 | 95.8±0.8 | 95.6± 2.0 |
| | HP | 99.6±7.5 | 93.6± 15.1 | 105.9±12.1 | 107.5± 2.9 |
| | WK | 111.3±0.1 | 111.4± 0.3 | 106.1±10.0 | 112.1± 0.4 |
| Med.-E | HC | 92.1±1.8 | 91.1± 3.4 | 92.1 ± 1.1 | 91.4± 2.5 |
| | HP | 74.6±19.3 | 78.4± 19.0 | 94.5±14.9 | 61.7± 29.2 |
| | WK | 109.3±0.5 | 109.9± 0.4 | 109.1±0.5 | 110.1± 0.7 |
| Med.-R | HC | 44.9±0.2 | 45.2± 0.8 | 44.7±0.5 | 46.7± 0.6 |
| | HP | 83.5±17.7 | 89.2± 8.1 | 83.2±17.2 | 98.5± 2.0 |
| | WK | 78.6±8.9 | 82.1± 3.8 | 77.8±3.5 | 86.7± 3.2 |
| Med. | HC | 46.4± 0.4 | 47.5± 0.4 | 46.5±0.3 | 48.6± 0.4 |
| | HP | 67.0±4.0 | 71.2± 6.5 | 66.2±5.5 | 86.4± 7.6 |
| | WK | 81.2±2.9 | 79.4± 4.7 | 81.6±2.0 | 85.0± 0.8 |
| Rand. | HC | 14.8±2.1 | 15.8± 1.6 | 12.2±1.1 | 16.3± 1.2 |
| | HP | 12.2±9.5 | 12.0± 10.0 | 15.7±10.0 | 20.4± 9.8 |
| | WK | 1.9±2.2 | 2.5± 2.6 | 2.3±1.8 | 2.3± 2.0 |
| | Total | 1013.3 | 1026.5 | 1033.7 | 1069.3 |

Table 14: Results on mujoco tasks using D4RL-v2 datasets and AntMaze tasks. Average normalized scores over the last 10 test episodes and five seeds are shown. HC = HalfCheetah, HP = Hopper, WK = Walker2d.

| | | AWAC | cV2AE | V2AE | infoV2AE |
|---|---|---|---|---|---|
| Expert | HC | 94.8± 0.2 | 95.8±0.8 | **97.0± 1.0** | 95.6± 2.0 |
| | HP | 109.8± 2.9 | **109.2±2.4** | 93.6± 15.1 | 107.5± 2.9 |
| | WK | 111.0± 0.2 | 102.1±12.7 | **111.4± 0.3** | **112.1± 0.4** |
| Med.-E | HC | 92.7± 0.8 | **92.4±3.4** | 91.1± 3.4 | 91.4± 2.5 |
| | HP | 98.6± 10.7 | 59.8±22.8 | 78.4± 19.0 | **94.5±14.9** |
| | WK | 109.2± 0.3 | 73.4±45.6 | 109.9± 0.4 | **110.1± 0.7** |
| Med.-R | HC | 40.9± 0.6 | 44.3±0.3 | 45.2± 0.8 | **46.7± 0.6** |
| | HP | 38.2± 9.4 | **98.9±2.3** | 89.2± 8.1 | **98.5± 2.0** |
| | WK | 65.0± 15.7 | **87.6±1.6** | 82.1± 3.8 | 86.7± 3.2 |
| Med. | HC | 44.3± 0.2 | 47.2±0.1 | 47.5± 0.4 | **48.6± 0.4** |
| | HP | 57.5± 3.0 | 75.0±5.8 | 71.2± 6.5 | **86.4± 7.6** |
| | WK | 81.0± 2.5 | 83.8±1.2 | 79.4± 4.7 | **85.0± 0.8** |
| Rand. | HC | 3.2± 1.3 | **20.6±0.6** | 15.8± 1.6 | 16.3± 1.2 |
| | HP | 7.3± 0.9 | 16.4±8.9 | 12.0± 10.0 | **20.4± 9.8** |
| | WK | 3.1± 1.0 | -0.2±0.1 | **2.5± 2.6** | **2.3± 2.0** |
| | Total | 956.5 | 1009.0 | 1026.5 | **1102.1** |

## J   THE CRITIC LOSS AND NORMALIZED SCORE IN MUJOCO TASKS

We show the critic loss and the normalized scores for MuJoCo tasks in Figures 6–8. The critic loss in V2AE is comparable to or lower than that of AWAC in MuJoCo tasks. However, there are exceptions, such as hopper-expert-v2 and hopper-medium-expert-v2, where the critic loss surges in V2AE but not in AWAC. Thus, it is necessary to be aware that the use of the mixture policy may compound the accumulation of the critic error in some rare cases, while the use of the mixture policy often mitigates the accumulation of the critic error for many tasks. Regarding the use of the continuous latent variable, while cV2AE often achieves the lower values of the critic loss than AWAC, the critic loss of cV2AE occasionally explodes. A possible explanation for these results may be that learning the continuous latent variable can provide flexibility to a policy representation but separate modes of the objective function is still interpolated in the learned latent space, which results in generating out-of-distribution samples.

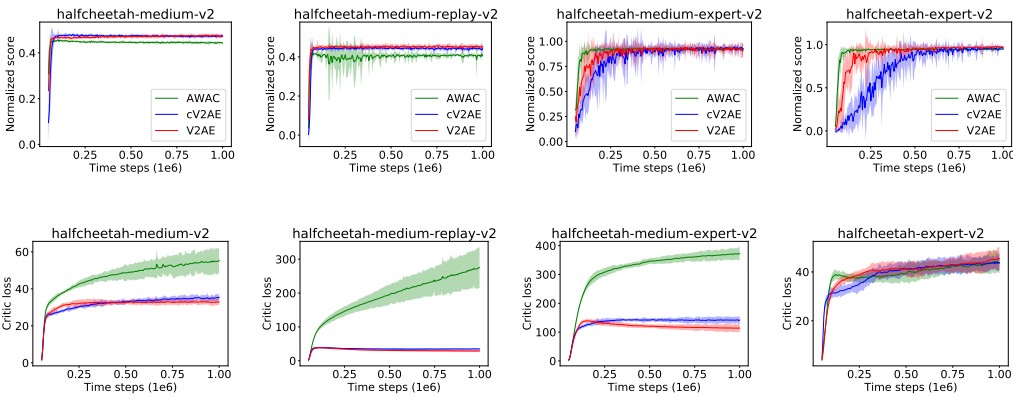

Figure 6: Critic loss and normalized score during training on HalfCheetah tasks.

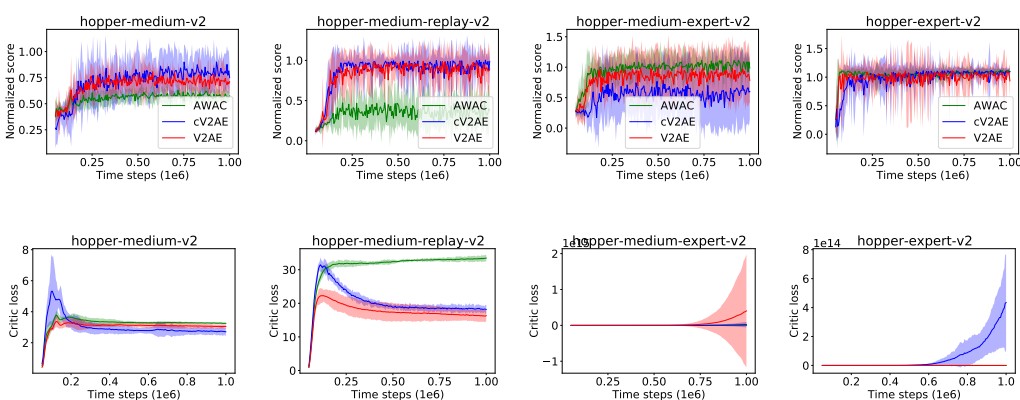

Figure 7: Critic loss and normalized score during training on hopper tasks.

## K ADDITIONAL RESULTS FOR THE ACTIVATION OF THE LATENT-CONDITIONED POLICY

Figure 9 shows the activation of sub-policies in V2AE on the pen-human-v0 task in three episodes. Here, we used the same policy trained with 10,000 updates. As shown, the target orientation of the object is different in each episode, and different sub-policies are activated to achieve the given goals. The sub-policy corresponding to $z = 7$ (yellow) is often activated during 20-40 steps. We

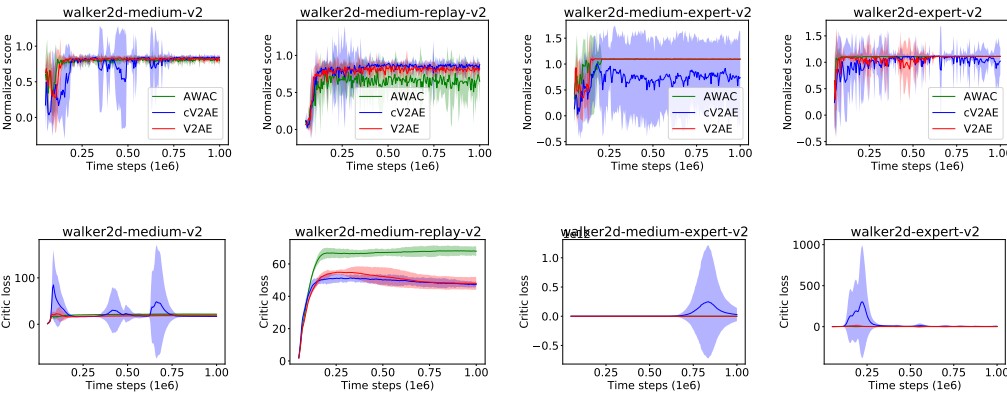

Figure 8: Critic loss and normalized score during training on walker2d tasks.

think that the sub-policy corresponding to $z = 7$ is effective for changing the orientation of the pen in this task. On the contrary, the sub-policies corresponding to $z = 0, 1, 2$ are not activated in many episodes. We think that these sub-policies correspond to the state-action pairs with low Q-values. In Offline RL, a given dataset often contains the mixed quality of the behavior, and it is not effective to reproduce all the behavior. Rather, the behaviors correspond to the low scores should be ignored. We think that the proposed algorithm can implicitly achieve this by dividing the state-action space and investigate the quality of the sub-policies corresponding to each region. This qualitative result supports the claim that the different behaviors are encoded in each of the sub-policies.

A visualization of the state-action pairs and the corresponding value of the latent variable on the kitchen-complete-v0 task is provided in Figure 10. We also show the histogram of the activated sub-policies in Figure 11 and the activation of sub-policies in a successful episode on the kitchen-complete-v0 task in Figure 12.

## L    HYPERPARAMETERS AND IMPLEMENTATION DETAILS

**Computational resource and license**    The experiments were run with Amazon Web Service and workstations with NVIDIA RTX 3090 GPU and Intel Core i9-10980XE CPU at 3.0GHz. We used the physics simulator MuJoCo (Todorov et al., 2012) under the institutional license, and later we switched to the Apache license.

**Software**    The software versions used in the experiments are listed below:

- Python 3.8
- Pytorch 1.10.0
- Gym 0.21.0
- MuJoCo 2.1.0
- mujoco-py 2.1.2.14

We used the author implementations for TD3+BC[2], CQL, and EDAC[3]. For mujoco-v2 and Antmaze tasks, we used the updated CQL implementation[4]. For mujoco-v0, Kitchen and Adroit tasks, we used the original CQL implementation[5]. For a fair comparison with V2AE, we implemented easy-BCQ and AWAC ourselves. For easyBCQ, we used the hyperparameters used in the author-provided implementation[6]. We implemented V2AE and AWAC based on the the author-provided implementation of TD3. In our implementation of easyBCQ and AWAC, double-clipped Q-learning was employed. To minimize the difference between V2AE and AWAC, we also used a delayed update of the policy in both V2AE and AWAC. For simplicity, we did not use a regularization technique for the actor such as the dropout layer used in (Kostrikov et al., 2022), although the use of such techniques should further improve performance. In our implementation of V2AE, the value of z is a part of the input to the actor network. Thus, the different behaviors corresponding to the different values of $z$ are represented by the same actor network.

**Computation of the advantage function**    In V2AE, a policy is deterministic, as both the gating policy $\pi(\boldsymbol{z}|\boldsymbol{s})$ and sub-policy $\pi(\boldsymbol{a}|\boldsymbol{s}, \boldsymbol{z})$ are deterministic. Thus, the state-value function is given by

$$V^{\pi}(\boldsymbol{s}) = \max_{\boldsymbol{z}} Q^{\pi}(\boldsymbol{s}, \boldsymbol{\mu}(\boldsymbol{s}, \boldsymbol{z})). \tag{41}$$

Therefore, the advantage function is given by

$$A^{\pi}(\boldsymbol{s}, \boldsymbol{a}) = Q^{\pi}(\boldsymbol{s}, \boldsymbol{a}) - V^{\pi}(\boldsymbol{s}) = Q^{\pi}(\boldsymbol{s}, \boldsymbol{a}) - \max_{\boldsymbol{z}} Q^{\pi}(\boldsymbol{s}, \boldsymbol{\mu}(\boldsymbol{s}, \boldsymbol{z})). \tag{42}$$

In the update of the policy, we used the target actor in the second term in equation 42. Thus, in our implementation, the advantage function is approximated with

$$A(\boldsymbol{s}, \boldsymbol{a}; \boldsymbol{w}, \boldsymbol{\theta}') = Q(\boldsymbol{s}, \boldsymbol{a}; \boldsymbol{w}) - \max_{\boldsymbol{z}} Q(\boldsymbol{s}, \boldsymbol{\mu}_{\boldsymbol{\theta}'}(\boldsymbol{s}, \boldsymbol{z}); \boldsymbol{w}). \tag{43}$$

---

[2]https://github.com/sfujim/TD3_BC
[3]https://github.com/snu-mllab/EDAC
[4]https://github.com/young-geng/CQL
[5]https://github.com/aviralkumar2907/CQL
[6]https://github.com/davidbrandfonbrener/onestep-rl

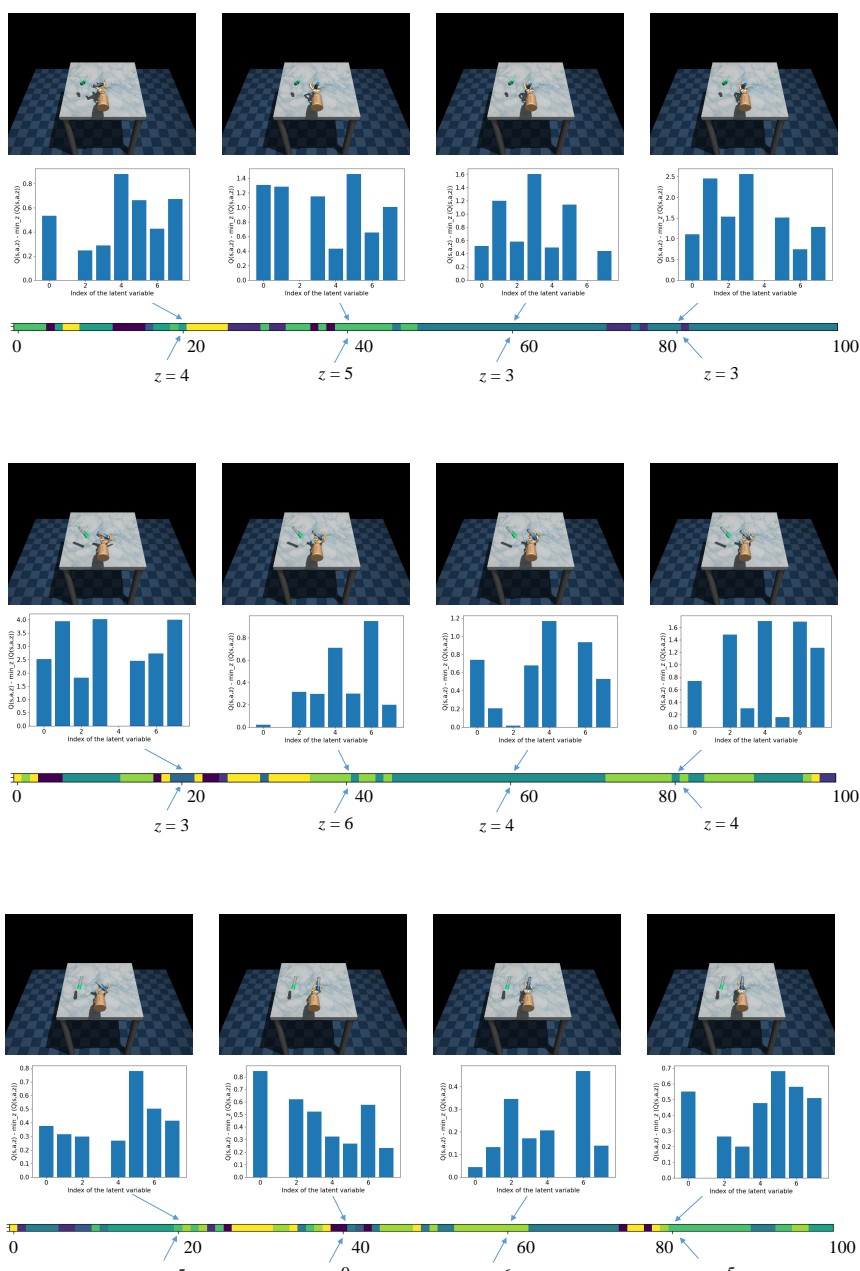

Figure 9: Visualization of activation of sub-policies in the pen-human-v0 task. The top row shows the state at the 20th, 40th, 60th, and 80th time steps; the graphs in the middle row show the action-values of each of the sub-policies at each state.

**Target smoothing in V2AE**    In V2AE, a policy is given by a mixture of deterministic sub-policies, where a sub-policy is selected in a deterministic manner, as in equation 3. Thus, the mixture policy in our framework is deterministic. As reported by Fujimoto & Gu (2021), the use of a deterministic policy may lead to an overfitting of the critic to the narrow peaks. Since our policy is deterministic, we also employed a technique called target policy smoothing used in TD3. Thus, the target value in

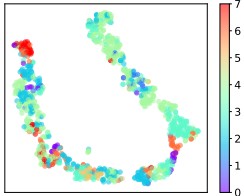 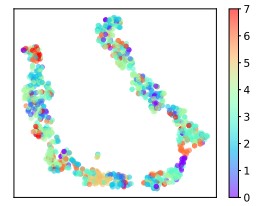 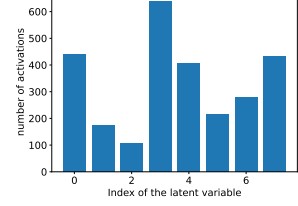

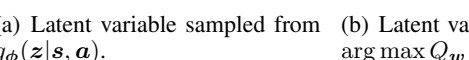

(a) Latent variable sampled from $q_{\boldsymbol{\phi}}(\boldsymbol{z}|\boldsymbol{s},\boldsymbol{a})$.

(b) Latent variable given by $\boldsymbol{z} = \arg\max Q_{\boldsymbol{w}}\big(\boldsymbol{s}, \boldsymbol{\mu}(\boldsymbol{s},\boldsymbol{z})\big)$.

Figure 10: Visualization of state-action pairs in the kitchen-complete-v0 task in D4RL. The color indicates the values of the latent variable.

Figure 11: Histogram of the activated latent variables on kitchen-complete-v0 in 10 episodes.

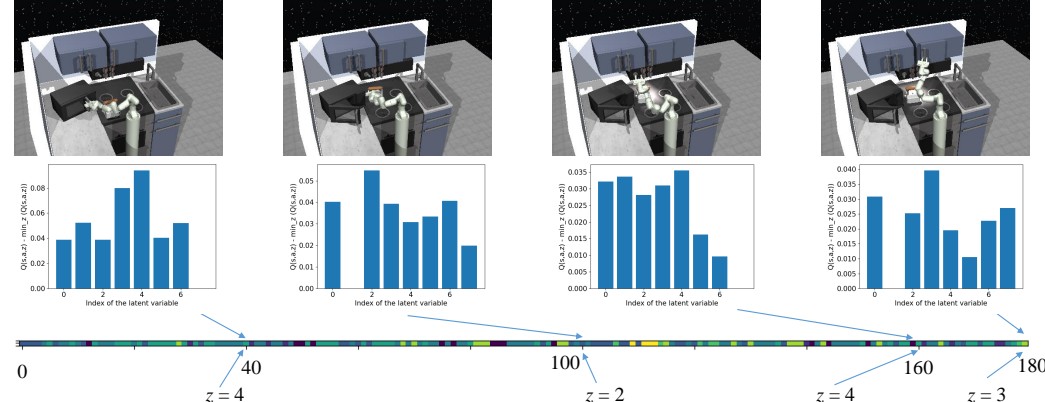

Figure 12: Visualization of the activation of sub-policies in a successful episode in the kitchen-complete-v0 task. The top row shows the state at the 40th, 100th, 160th, and 180th time steps; the graphs in the middle row show the action-values of each of the sub-policies at each state.

equation 14 is modified as

$$y_i = r_i + \gamma \max_{\boldsymbol{z}' \in \mathcal{Z}} \min_{j=1,2} Q_{\boldsymbol{w}'_j}\big(\boldsymbol{s}', \boldsymbol{\mu}_{\boldsymbol{\theta}'}(\boldsymbol{s}', \boldsymbol{z}')\big) + \epsilon_{\text{clip}}), \tag{44}$$

where $\epsilon_{\text{clip}}$ is given by

$$\epsilon_{\text{clip}} = \min(\max(\epsilon, -c), c) \quad \text{where} \quad \epsilon \sim \mathcal{N}(0, \sigma), \tag{45}$$

and the constant $c$ defines the range of the noise.

**Mutual-information-based regularization**    To implement the MI-based regularization, we introduced another network to represent $g_{\boldsymbol{\psi}}(\boldsymbol{z}|\boldsymbol{s},\boldsymbol{a})$ in addition to a network that represent the posterior distribution $q_{\boldsymbol{\phi}}(\boldsymbol{z}|\boldsymbol{s},\boldsymbol{a})$. When maximizing the objective $\mathcal{L}_{\text{ML}}$ in equation 9, both the actor $\mu_{\boldsymbol{\theta}}(s,z)$ and the auxiliary distribution $g_{\boldsymbol{\psi}}(\boldsymbol{z}|\boldsymbol{s},\boldsymbol{a})$ are updated, but the posterior distribution $q_{\boldsymbol{\phi}}(\boldsymbol{z}|\boldsymbol{s},\boldsymbol{a})$ is frozen. When maximizing $\sum_{i=1}^{N} \mathbb{E}_{\boldsymbol{z} \sim p(\boldsymbol{z})} \log g_{\boldsymbol{\psi}}(\boldsymbol{z}|\boldsymbol{s}_i, \boldsymbol{\mu}_{\boldsymbol{\theta}}(\boldsymbol{s}_i, \boldsymbol{z}))$, both the actor $\mu_{\boldsymbol{\theta}}(s,z)$ and the posterior distribution $q_{\boldsymbol{\phi}}(\boldsymbol{z}|\boldsymbol{s},\boldsymbol{a})$ are updated, but the auxiliary distribution $g_{\boldsymbol{\psi}}(\boldsymbol{z}|\boldsymbol{s},\boldsymbol{a})$ is frozen. For maximizing $\log g_{\boldsymbol{\psi}}(\boldsymbol{z}|\boldsymbol{s}_i, \boldsymbol{\mu}_{\boldsymbol{\theta}}(\boldsymbol{s}_i, \boldsymbol{z}))$, the latent variable is sampled from the prior distribution, i.e. the uniform distribution in this case, and maximization of $\log g_{\boldsymbol{\psi}}(\boldsymbol{z}|\boldsymbol{s}_i, \boldsymbol{\mu}_{\boldsymbol{\theta}}(\boldsymbol{s}_i, \boldsymbol{z}))$ is approximated as minimizing the squared difference $||\boldsymbol{z} - \hat{\boldsymbol{z}}||^2$, where $\hat{\boldsymbol{z}}$ is the output of the network that represents $g_{\boldsymbol{\psi}}(\boldsymbol{z}|\boldsymbol{s},\boldsymbol{a})$.

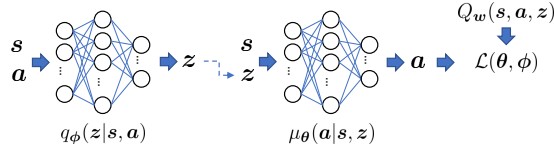

(a) Computation for maximizing $\mathcal{L}_{\mathrm{ML}}$ in equation 9.

(b) Computation for maximizing $\sum_{i=1}^{N} \mathbb{E}_{\boldsymbol{z} \sim p(\boldsymbol{z})} \log g_{\boldsymbol{\psi}}(\boldsymbol{z}|\boldsymbol{s}_i, \boldsymbol{\mu}_{\boldsymbol{\theta}}(\boldsymbol{s}_i, \boldsymbol{z}))$.

Figure 13: Connection between $q_{\phi}(\boldsymbol{z}|\boldsymbol{s}, \boldsymbol{a})$, $\mu_{\boldsymbol{\theta}}(\boldsymbol{s}, \boldsymbol{z})$, and $g_{\boldsymbol{\psi}}(\boldsymbol{z}|\boldsymbol{s}, \boldsymbol{a})$ during the training.

**Implementation of a variant of V2AE with the continuous latent variable**  We refer to a variant of V2AE with the continuous latent variable as cV2AE in this paper. The differences from our main algorithm V2AE, which employs the discrete latent variable, are the reparametrization of sampling the latent variable, the KL divergence term in the loss function, and how to compute $\boldsymbol{z}^* = \arg\max_{\boldsymbol{z}} Q_{\boldsymbol{w}}(\boldsymbol{s}, \boldsymbol{\mu}(\boldsymbol{s}, \boldsymbol{z}))$. We assume that the prior distribution $p(\boldsymbol{z})$ and posterior distribution $p(\boldsymbol{z}|\boldsymbol{s}, \boldsymbol{a})$ is Gaussian, and we used the reparameterization as in the standard VAE as

$$\boldsymbol{z} = \boldsymbol{\mu}_z + \boldsymbol{\epsilon} \cdot \boldsymbol{\sigma}_z, \tag{46}$$

where $\boldsymbol{\mu}_z$ and $\boldsymbol{\sigma}_z$ are the mean and the standard deviation of the posterior distribution $p(\boldsymbol{z}|\boldsymbol{s}, \boldsymbol{a})$. As the latent variable $\boldsymbol{z}$ is continuous, we cannot compute the softmax distribution in equation 8 given by

$$p(\boldsymbol{z}|\boldsymbol{s}) = \frac{\exp\left(Q_{\boldsymbol{w}}(\boldsymbol{s}, \boldsymbol{\mu}_{\boldsymbol{\theta}}(\boldsymbol{s}, \boldsymbol{z}))\right)}{\sum_{\boldsymbol{z} \in \mathcal{Z}} \exp\left(Q_{\boldsymbol{w}}(\boldsymbol{s}, \boldsymbol{\mu}_{\boldsymbol{\theta}}(\boldsymbol{s}, \boldsymbol{z}))\right)}. \tag{47}$$

Thus, we assumed that $p(\boldsymbol{z}|\boldsymbol{s})$ as the Gaussian distribution as in previous studies (Fujimoto et al., 2019), and the KL divergence term in the objective function in equation 7 is computed as in the standard VAE (Kingma & Welling, 2014). When the latent variable $\boldsymbol{z}$ is continuous, we cannot explicitly compute $\boldsymbol{z}^* = \arg\max_{\boldsymbol{z}} Q_{\boldsymbol{w}}(\boldsymbol{s}, \boldsymbol{\mu}(\boldsymbol{s}, \boldsymbol{z}))$. Thus, we approximated it as follows: we generate $N$ samples as $\boldsymbol{z}_i \sim \mathcal{N}(0, 1)$ for $i = 0, \ldots, N$ and determine $\hat{\boldsymbol{z}}^* = \arg\max_{\boldsymbol{z}_i} Q_{\boldsymbol{w}}(\boldsymbol{s}, \boldsymbol{\mu}(\boldsymbol{s}, \boldsymbol{z}_i))$. In our implementation, the continuous latent variable $\boldsymbol{z}$ is two-dimensional, and $N = 100$. When the dimensionality of $\boldsymbol{z}$ is higher, the approximation of $\boldsymbol{z}^* = \arg\max_{\boldsymbol{z}} Q_{\boldsymbol{w}}(\boldsymbol{s}, \boldsymbol{\mu}(\boldsymbol{s}, \boldsymbol{z}))$ gets more coarse. We found that the performance is significantly lower when the dimensionality of $\boldsymbol{z}$ is more than two.

**Number of updates**  In kitchen-complete-v0, pen-human-v0, hammer-human-v0, door-human-v0, and relocate-human-v0, the number of samples contained in the dataset is significantly smaller than those for other datasets. While the datasets for MuJoCo tasks contained approximately 1 million samples, the numbers of samples in the Adroit-human tasks and the kitchen-complete-v0 were as follows: kitchen-complete-v0: 3679 samples; pen-human-v0: 4950 samples; hammer-human-v0: 11264 samples; door-human-v0: 6703 samples; and relocate-human-v0: 9906 samples. Thus, in kitchen-complete-v0, pen-human-v0, hammer-human-v0, door-human-v0, and relocate-human-v0, we updated the policy 10,000 times, while for the other tasks we updated the policy 1 million times. The above-mentioned number of policy updates was applied to all methods except easyBCQ, which is a one-step offline RL algorithm. For easyBCQ, the behavior policy was trained 500,000 times and the critics were trained 2 million times, except for *-human-v0 tasks in Adroit tasks, where the behavior policy was trained 50,000 times, and the critics were trained 200,000 times.

**Hyperparameters**  Tables 15–19 provide the hyperparameters used in the experiments.

Table 15: Hyperparameters of V2AE&infoV2AE.

|  | Hyperparameter | Value |
|---|---|---|
| Hyperparameters | Optimizer | Adam |
|  | Critic learning rate | 3e-4 |
|  | Actor learning rate | 3e-4 |
|  | Posterior learning rate | 3e-4 |
|  | Mini-batch size | 256 |
|  | Discount factor | 0.99 |
|  | Target update rate | 5e-3 |
|  | Policy noise | 0.2 |
|  | Policy noise clipping | (-0.5, 0.5) |
|  | Policy update frequency | 2 |
| Architecture | Critic hidden dim | 256 |
|  | Critic hidden layers | 2 |
|  | Critic activation function | ReLU |
|  | Actor hidden dim | 256 |
|  | Actor hidden layers | 2 |
|  | Actor activation function | ReLU |
|  | Posterior hidden dim | 256 |
|  | Posterior hidden layers | 2 |
|  | Posterior activation function | ReLU |
| V2AE | Score scaling $\alpha$ | 5.0 (kitchen, human, Antmaze) |
|  |  | 10.0 (mujoco-v2, mujoco-v1) |
| infoV2AE | learning rate of the posterior for infomax | 3e-6 (Kitchen, Adroit) |
|  |  | 5e-7 (mujoco-v2) |
|  |  | 5e-7 (Antmaze) |
|  | Score scaling $\alpha$ | 5.0 (Antmaze, hopper-medium-expert) |
|  |  | 10.0 (others) |

Table 16: Hyperparameters of AWAC.

|  | Hyperparameter | Value |
|---|---|---|
| Hyperparameters | Optimizer | Adam |
|  | Critic learning rate | 3e-4 |
|  | Actor learning rate | 3e-4 |
|  | Mini-batch size | 1024 |
|  | Discount factor | 0.99 |
|  | Target update rate | 5e-3 |
|  | Policy update frequency | 2 |
|  | Score scaling $\alpha$ | 10.0 |
| Architecture | Critic hidden dim | 256 |
|  | Critic hidden layers | 2 |
|  | Critic activation function | ReLU |
|  | Actor hidden dim | 256 |
|  | Actor hidden layers | 2 |
|  | Actor activation function | ReLU |

Table 17: Hyperparameters of TD3+BC. The default hyperparameters in the TD3+BC GitHub are used.

|  | Hyperparameter | Value |
| --- | --- | --- |
| | Optimizer | Adam |
| | Critic learning rate | 3e-4 |
| | Actor learning rate | 3e-4 |
| | Mini-batch size | 256 |
| TD3 Hyperparameters | Discount factor | 0.99 |
| | Target update rate | 5e-3 |
| | Policy noise | 0.2 |
| | Policy noise clipping | (-0.5, 0.5) |
| | Policy update frequency | 2 |
| | Critic hidden dim | 256 |
| | Critic hidden layers | 2 |
| Architecture | Critic activation function | ReLU |
| | Actor hidden dim | 256 |
| | Actor hidden layers | 2 |
| | Actor activation function | ReLU |
| TD3+BC Hyperparameters | $\alpha$ | 2.5 |

Table 18: Hyperparameters of CQL. The default hyperparameters in the CQL GitHub are used.

|  | Hyperparameter | Value |
| --- | --- | --- |
| | Optimizer | Adam |
| | Critic learning rate | 3e-4 |
| | Actor learning rate | 3e-5 |
| | Mini-batch size | 256 |
| SAC Hyperparameters | Discount factor | 0.99 |
| | Target update rate | 5e-3 |
| | Target entropy | -1·Action Dim |
| | Entropy in Q target | False |
| | Critic hidden dim | 256 |
| | Critic hidden layers | 3 |
| Architecture | Critic activation function | ReLU |
| | Actor hidden dim | 256 |
| | Actor hidden layers | 3 |
| | Actor activation function | ReLU |
| | Lagrange | False |
| | $\alpha$ | 10 |
| CQL Hyperparameters | Pre-training steps | 40e3 |
| | Num sampled actions (during eval) | 10 |
| | Num sampled actions (logsumexp) | 10 |

Table 19: Hyperparameters of easyBCQ. The default hyperparameters in the easyBCQ in the author-provided codes in GitHub are used.

|  | Hyperparameter | Value |
| --- | --- | --- |
| | Optimizer | Adam |
| | Critic learning rate | 1e-4 |
| | Behavior policy learning rate | 1e-4 |
| | Mini-batch size | 512 |
| easyBCQ Hyperparameters | Discount factor | 0.99 |
| | Target update rate | 5e-3 |
| | number of action samples | 100 |
| | Critic hidden dim | 1024 |
| | Critic hidden layers | 2 |
| Architecture | Critic activation function | ReLU |
| | Behavior policy hidden dim | 1024 |
| | Behavior policy hidden layers | 2 |
| | Behavior policy activation function | ReLU |

