# OpenReview forum: "On the Importance of the Policy Structure in Offline Reinforcement Learning"
_ICLR.cc/2023/Conference — Submitted to ICLR 2023_

### Official Review · Reviewer_chjv · 2022-10-23

**Confidence:** 3
**Correctness:** 3
**Technical Novelty And Significance:** 3
**Empirical Novelty And Significance:** 3
**Recommendation:** 6

**Clarity, Quality, Novelty And Reproducibility:**


The individual techniques used in this paper are not novel (e.g. VAE, modeling clusters of data, MI, softmax distribution given by the Q values) but seeing them combined together in the context of offline RL is.  This aspect makes this work original.

The paper would benefit from some further clarifications. For example, some details in the equation 7 is not immediately obvious (e.g. why is the expected value in the second term over p(z) not p(z|s_i)?)  and would merit a derivation in the appendix.  Another example is eq 10. I understand how the equation follows from Barber & Agakov (2003) and for the need of an auxiliary distribution. But how is g(z|s,a) connected with q(z|s,a)? Similarly in Algorithm 1, is the actor and the posterior updated in the same gradient step?

Another clarification question I have is whether for each value of z, you have a different actor network, or do they all share the same network, and the value of z is just a part of the input to the same actor network? Afaiu, it is the latter. Let me know if I am wrong.



**Strength And Weaknesses:**


The paper pointed out an often ignored factor of offline RL, which is that the dataset may come from multiple behavior policies and forcing one policy to adapt to all of them at once may make learning harder. The proposed solution is mathematically grounded and the motivation is clearly shown in the toy problem. The authors also illustrated the connection between policy structure and the critic loss. All of those points above make this paper a strong candidate.

There are also some weaknesses of the paper. One is that it is unclear what exactly caused fitting the multimodal distribution hard and whether it is a fundamental problem or something that will automatically go away with more advanced network architecture with stronger regularization, without explicitly modeling the latent z. Another area that can be improved is the clarity. (see the next section)


**Summary Of The Paper:**

This work aims to draw attention to the issue that there may be multiple “modalities” to the behavior data collected for offline RL and it is hypothesized that it is harmful to force a single policy to learn from the data from all those different behavior policies.

The paper proposed to use V2AE (a VAE inspired method) to separate out the different modalities in the behavior data and use one policy for each of those modalities. They propose a gating policy that selects the latent modality z with the maximum Q value, which in practice is approximated with a softmax distribution inthe variational lower bound. In addition, the authors found that by adding a MI regularization term that encourages diversity of the latent modality per (s,a) pair, they got improved performance.

In terms of the experiment results, the authors found that their proposal does well on a toy problem where there are two potential goals that the agents can head to. Existing methods exhibits hesitation around forks in the path towards those two goals, whereas the V2AE method solves it perfectly. More encouragingly, such gain is still exists for AntMaze, where the problem setup can be seen as a more complicated version of the toy problem.


**Summary Of The Review:**

I recommend a score of 6 because this paper pointed out a unique perspective to the failure modes of offline RL methods and proposed a preliminary approach that combines various other areas of ML to solve it. It is valuable due to the novel perspective and to the sound motivation as demonstrated in the toy problem. Whether there are easier solutions to the current approach is still yet to be seen, and the clarity can be improved. That is why I did not give a higher score.

---

> ### Author Response · Authors · 2022-11-11
> **Response to Reviewer chjv**
>
> Thank you for evaluating our paper and giving constructive feedback to us. We summarize our response to the comments and our revisions.
>
>   >some details in the equation 7 is not immediately obvious (e.g. why is the expected value in the second term over $p(z)$ not $p(z|s_i)$?) and would merit a derivation in the appendix.
>
> Thank you for the question. In equation 7, the second term should be actually the expectation with respect to $p(z|s_i,a_i)$. In the revised version, we provide the derivation of the lower bound in the appendix.
>
> >Another example is eq 10. I understand how the equation follows from Barber & Agakov (2003) and for the need of an auxiliary distribution. But how is g(z|s,a) connected with q(z|s,a)? Similarly in Algorithm 1, is the actor and the posterior updated in the same gradient step?
>
> We introduce a network to represent $g(z|s,a)$ in addition to a network that represent the posterior distribution $q(z|s,a)$. When updating $g(z|s,a)$, both the actor $\mu(s,z)$ and the auxiliary distribution $g(z|s,a)$ are updated, but the posterior distribution $q(z|s,a)$ is frozen. When updating $q(z|s,a)$, both the actor $\mu(s,z)$ and the posterior distribution $q(z|s,a)$ are updated, but the auxiliary distribution $g(z|s,a)$ is frozen. To clarify this point, we added detailed descriptions in the appendix.  In addition, we also modified Algorithm 1 to clearly indicate that the weighted ML term and the mutual information term are separately updated.
>
> >Another clarification question I have is whether for each value of z, you have a different actor network, or do they all share the same network, and the value of z is just a part of the input to the same actor network? Afaiu, it is the latter. Let me know if I am wrong.
>
> As the reviewer understands, the value of z is a part of the input to the actor network. Thus, the different behaviors corresponding to the different values of $z$ are represented by the same actor network.  We will clearly describe this in the revised version.

---

> > ### Author Response · Authors · 2022-11-18
> > **Any response will be appreciated**
> >
> > Dear Reviewer chjv,
> >
> > After the initial revision, we updated the manuscript by adding baselines requested by other reviewers. We believe that the empirical results are now stronger than the initial submission. We would highly appreciate it if the reviewer gives us any feedback.
> >
> > Best regards,
> > Authors

---

### Official Review · Reviewer_NNyS · 2022-10-24

**Confidence:** 3
**Clarity, Quality, Novelty And Reproducibility:** In general the paper is easy to follo…
**Correctness:** 3
**Technical Novelty And Significance:** 2
**Empirical Novelty And Significance:** 2
**Recommendation:** 3

**Strength And Weaknesses:**

# Strength
* More expressive policies are an interesting direction of offline reinforcement learning research.

# Weaknesses
* The approach lacks novelty. A VAE-style policy learning scheme has already been explored in [1]. While [2] proposes more expressive autoregressive policies.
* The paper lacks comparisons to the current SOTA results. In particular, [1] and [3] achieve significantly stronger results on the antmaze dataset, which are considered to be more challenging for offline reinforcement learning [4].
* The advantage scaling term (15) relies on the minimum and maximum over minibatches that might be sensitive to outliers.

[1] Latent-Variable Advantage-Weighted Policy Optimization for Offline RL
Xi Chen, Ali Ghadirzadeh, Tianhe Yu, Yuan Gao, Jianhao Wang, Wenzhe Li, Bin Liang, Chelsea Finn, Chongjie Zhang

[2] EMaQ: Expected-Max Q-Learning Operator for Simple Yet Effective Offline and Online RL
Seyed Kamyar Seyed Ghasemipour, Dale Schuurmans, Shixiang Shane Gu

[3] Offline Reinforcement Learning with Implicit Q-Learning
Ilya Kostrikov, Ashvin Nair, Sergey Levine

[4] RvS: What is Essential for Offline RL via Supervised Learning?
Scott Emmons, Benjamin Eysenbach, Ilya Kostrikov, Sergey Levine

**Summary Of The Paper:**

The paper proposes a more expressive policy for offline reinforcement learning. In particular, the paper proposes to use a mixture of deterministic policies.

**Summary Of The Review:**

The paper lacks novelty and omits comparisons to the current SOTA. For these reasons, I believe that they paper falls below the acceptance threshold.

---

> ### Author Response · Authors · 2022-11-11
> **Reponse to Reviewer NNyS**
>
> >The paper lacks comparisons to the current SOTA results. In particular, [1] and [3] achieve significantly stronger results on the antmaze dataset, which are considered to be more challenging for offline reinforcement learning [4].
>
> Thank you for the suggestion. We will provide a comparison with LAPO proposed in [1], which will strengthen our results.
> For now, we can say that V2AE outperforms LAPO and IQL on mujoco tasks. While the performance of V2AE on Antmaze tasks in our experiments are lower than that of LAPO reported in [1], we found that the author's implementation of LAPO includes a few techniques to improve the performance, e.g. scaling the reward, clipping the target value for the state-value function, adaptive combination of double critics ($Q_{target}=0.7 \min(Q_1, Q_2) + 0.3\max(Q_1, Q_2)$). Although these techniques could be used in TD3+BC, AWAC, and V2AE, they are not incorporated into our results.
> For a fair comparison with our results, we evaluated LAPO without these techniques, which we refer to as LAPO- in the table below.
> We show the results over 5 random seeds with 100 test episodes.
>
> |       | LAPO- | V2AE | infoV2AE |
> | ---- | ---- | ---- | ---- |
> | antmaze-umaze-v0 | 76.4±3.3 | 83.6 $\pm$4.5 | 88.4$\pm$3.6 |
> | antmaze-umaze-diverse-v0 | 56.6±8.6 | 43.2$\pm$ 7.8 | 52.8$\pm$7.9 |
> | antmaze-medium-play-v0 | 0.8±1.6 | 77.0$\pm$ 5.1 | 48.6$\pm$25.3 |
> | antmaze-medium-diverse-v0 | 0.0±0.0 | 56.8$\pm$ 27.2 | 59.2$\pm$29.4
> | antmaze-large-play-v0 | 0.0±0.0 | 1.0$\pm$1.3 | 5.2 $\pm$8.4 |
> | antmaze-large-diverse-v0 | 0.0±0.0 | 4.8$\pm$9.6 | 6.6 $\pm$5.2 |
>
> The results indicate that the proposed method V2AE and infoV2AE outperform LAPO on Antmaze tasks, in the absence of techniques such as scaling the reward, clipping the target value for the state-value function, and adaptive combination of double critics ($Q_{target}=0.7 \min(Q_1, Q_2) + 0.3\max(Q_1, Q_2)$).
> Currently, we are running experiments for evaluating the proposed method combined with these techniques. We will report the results when finished.
>
> Regarding IQL in [3], we included the comparison with IQL and other baseline methods on mujoco-v2 tasks in the appendix, and V2AE outperformed the IQL on the locomotion tasks in D4RL. The reason why we excluded IQL from the main text is that the research direction of IQL is orthogonal to ours; IQL is the technique to train the critic, and our focus is the policy structure. In the main manuscrips, we focused on the baseline methods that use the double-clipped Q-learning in the critic to investigate the effect of the policy structure.
>
> In the current experiment, we did not use some techqniues to further improve the performance for simplicity. As we are evaluating V2AE with techniques used in LAPO, we report the comparison with IQL in Antmaze tasks  together with the comparison with LAPO.
>
> >The advantage scaling term (15) relies on the minimum and maximum over mini-batches that might be sensitive to outliers.
>
> Thank you for the comment. As the reviewer indicated, the minimum and maximum over mini-batches might be sensitive to outliers. However, we did not observe instability during the training phase. Thus, we think that the technique is not problematic in practice.
>
> >The approach lacks novelty. A VAE-style policy learning scheme has already been explored in [1]. While [2] proposes more expressive autoregressive policies.
>
> We acknowledge that several studies explored VAE-style policy learning. However, our contribution is the importance of the policy structure based on the discrete latent variable. Although the study [1] explored how to utilize the latent action space and the study [2] investigated how to leverage generative models in offline RL, they do not point out the benefit of the discrete latent variable. The benefit of the proposed method will be more apparent when compared with LAPO in [1].
>
> To address the comment from Reviewer NNys, we ran experiments with LAPO-, which does not have the target value clipping. Then, we found that the value of the critic loss function often exploded in the training phase of LAPO-. LAPO learns a policy in a continuous latent space, and as a result of interpolation between separate modes in the dataset, LAPO generates the OOD actions, as in cV2AE in our study. However, when we leverage the discrete latent variable as in our method, the problem of generating OOD actions is mitigated, and the accumulation of the critic loss value is reduced.
> To support the claim, we are currently preparing the results to show the critic loss value in LAPO-. We will report the results when the experiments are finished.
>
> We believe that these research directions are not fully explored and that our findings can give new insights to the community.

---

> ### Author Response · Authors · 2022-11-17
> **Comparison with LAPO**
>
> As a follow-up, we provide additional results in comparison with LAPO.
> As we mentioned in our previous comment,  we found that the author's implementation of LAPO includes a few techniques to improve performance.
> Such techniques are compatible with V2AE and our baseline methods, but they are not incorporated into our experiments.
> For this reason, we evaluated LAPO without these techniques, which we refer to as LAPO-.
>
> Overall, V2AE and infoV2AE outperformed LAPO- on mujoco-v2 tasks and Antmaze-v0 tasks.
> While our method V2AE leverages the discrete latent space, LAPO uses the continuous latent space.
> In this sense, LAPO is closer to cV2AE, which is a variant of V2AE that uses the continuous latent space.
> In the main manuscript, we showed that the critic loss value often explodes during the training of cV2AE.
> Similarly, the critic log value increases rapidly at the beginning of the training in LAPO-. (Please refer to Figure 5 in Appendix G.2)
> While the critic log value often decreases at the end of the training of LAPO-, the critic loss value is still higher than that of V2AE.
> As the performance of the policy is better in V2AE in these tasks, the surge of the critic loss value indicates the generation of the out-of-distribution (OOD) actions during the training in LAPO.
> These results support our observations in the main manuscript and indicate that the use of the discrete latent variable is effective to reduce the generation of OOD actions.
>
> |  | LAPO- | V2AE | infoV2AE |
> | --- | --- | --- | --- |
> | HalfCheetah-expert | 94.9$\pm$0.2 | **97.0$\pm$ 1.0** | 95.6$\pm$ 2.0 |
> | Hopper-expert | 111.4$\pm$0.6 | 93.6$\pm$ 15.1 | 107.5$\pm$ 2.9 |
> | Walker2d-expert | 111.1$\pm$0.2 | 111.4$\pm$ 0.3 | **112.1$\pm$ 0.4** |
> | HalfCheetah-medium-expert |  94.2$\pm$0.4 | 91.1$\pm$ 3.4 | 91.4$\pm$ 2.5 |
> | Hopper-medium-expert | 110.9$\pm$1.0 | 78.4$\pm$ 19.0 | 94.5$\pm$14.9 |
> | Walker2d-medium-expert | 110.4$\pm$0.1 | 109.9$\pm$ 0.4 | 110.1$\pm$ 0.7 |
> | HalfCheetah-medium-replay |  40.7$\pm$1.9 | **45.2$\pm$ 0.8** | **46.7$\pm$ 0.6** |
> | Hopper-medium-replay | 52.6$\pm$11.3 | **89.2$\pm$ 8.1** | **98.5$\pm$ 2.0** |
> | Walker2d-medium-replay | 60.1$\pm$11.7 | **82.1$\pm$ 3.8** | **86.7$\pm$ 3.2** |
> | HalfCheetah-medium | 45.5$\pm$0.2 | **47.5$\pm$ 0.4** | **48.6$\pm$ 0.4** |
> | Hopper-medium |  51.8$\pm$3.6 | **71.2$\pm$ 6.5** | **86.4$\pm$ 7.6** |
> | Walker2d-medium |  77.7$\pm$3.5 | 79.4$\pm$ 4.7 | **85.0$\pm$ 0.8** |
> | HalfCheetah-random |  2.3$\pm$0.3 | **15.8$\pm$ 1.6** | **16.3$\pm$ 1.2** |
> | Hopper-random | 21.8$\pm$11.3 | 12.0$\pm$ 10.0 | 20.4$\pm$ 9.8 |
> | Walker2d-random | 5.6$\pm$3.1 | 2.5$\pm$ 2.6 | 2.3$\pm$ 2.0 |
> | Antmaze-umaze-v0 | 76.4$\pm$3.3 | **83.6$\pm$4.5** | **88.4$\pm$3.6** |
> | Antmaze-umaze-diverse-v0  | 56.6$\pm$8.6 | 43.2$\pm$ 7.8 | 52.8$\pm$7.9 |
> | Antmaze-medium-play-v0 | 0.8$\pm$1.6 | **77.0$\pm$5.1** | **48.6$\pm$25.3** |
> | Antmaze-medium-diverse-v0 | 0.0$\pm$0.0 | **56.8$\pm$27.2** | **59.2$\pm$29.4** |
> | Antmaze-large-play-v0 | 0.0$\pm$0.0 | 1.0$\pm$1.3 | **5.2$\pm$ 8.4** |
> | Antmaze-large-diverse-v0 | 0.0$\pm$0.0 | **4.8$\pm$9.6** | **6.6$\pm$ 5.2** |
>
> The table above shows the results across five random seeds after 1 million updates. We used 10 test episodes for each random seed in mujoco-v2 locomotion tasks, and 100 test episodes for each random seed for Antmaze tasks.
> Bold texts indicate the results where V2AE and infoV2AE clearly outperformed LAPO-.
>
> We hope that this result addresses the reviewer's concern regarding the comparison with LAPO. If the reviewer thinks that any additional results are necessary, please let us know.

---

> ### Author Response · Authors · 2022-11-17
> **Comparison with IQL and LAPO on Antmaze**
>
> To address the concern raised by the reviewer, we provide the complete results on Antmaze. As we mentioned in previous comments, a few techniques to improve performance on Antmaze is incorporated in the authors' implementation of LAPO.
> As these techniques are not incorporated in our initial experiments, we performed additional experiments for fair comparison.
> In methods named like "xx_lapo", the techniques used in LAPO is incorporated.
> We summarized the techniques/differences as below:
> - computation of the target value as $Q_{target} = 0.7 \min (Q_1, Q_2) + 0.3 \max (Q_1, Q_2)$
> - target value clipping $v_{target} =  \min(\max(v_{target}, v_{min}), v_{max})$
> - reward scaling $r \leftarrow r\times 100$
> - network architecture: three hidden layers with 256 units, learning rate: 2e-4
>
> In LAPO-, IQL, V2AE, infoV2AE, we used the reward $r \leftarrow r - 1$, following the protocol used in the original IQL paper.
>
> While action normalization is used in LAPO, we found that the action normalization does not improve the performance of IQL and V2AE.
> Therefore, action normalization is used only in LAPO in our results.
> In the table below, LAPO- shows LAPO without these techniques, although the network architecture is the same as the original LAPO.
> "IQL (re-run)" indicates the results of re-running IQL in our code base, which fairly reproduced the results reported in the original IQL paper.
> "V2AE" shows the result reported in our main manuscript.
>
> |  | LAPO-  | LAPO (re-run)  | IQL(re-run) | IQL_lapo | V2AE | V2AE_lapo | infoV2AE | infoV2AE_lapo |
> | --- | --- | --- | --- |--- | --- |--- | --- |--- |
> | antmaze-umaze-v0 | 76.4$\pm$3.3 | **97.8$\pm$1.3** | 87.4$\pm$4.5  |  87.2$\pm$2.8 | 83.6 $\pm$4.5 | 92.8$\pm$2.1 | 88.4±3.6 | 89.4±5.1 |
> | antmaze-umaze-diverse-v0 | 56.6$\pm$8.6 |  52.4$\pm$20.5 | **64.6$\pm$5.6** | 54.0$\pm$13.4 | 43.2$\pm$ 7.8 | 32.6$\pm$25.8 | 52.8±7.9 | 34.8±18.0 |
> | antmaze-medium-play-v0 | 0.8$\pm$1.6 | 64.2$\pm$5.3 | **74.6$\pm$3.1** | 64.0$\pm$9.5 | **77.0$\pm$ 5.1** | 63.0$\pm$13.0 | 48.6±25.3 | 62.6±6.8 |
> | antmaze-medium-diverse-v0 | 0.0$\pm$0.0 | 69.4$\pm$12.8 | 73.8$\pm$7.1 | 54.8$\pm$8.6 | 56.8$\pm$ 27.2 | 75.0$\pm$8.5 | 59.2±29.4 | **82.8±4.4** |
> | antmaze-large-play-v0 | 0.0$\pm$0.0 | 12.4$\pm$16.9 | 39.0$\pm$7.2 | 20.4$\pm$11.0 | 1.0$\pm$ 1.3 | **42.2$\pm$23.0** | 5.2 ± 8.4 | **47.4±14.5** |
> | antmaze-large-diverse-v0 | 0.0$\pm$0.0 | 18.2$\pm$16.1 | 48.0$\pm$9.0 |  17.0$\pm$4.2 | 4.8$\pm$ 9.6 | **56.6$\pm$4.5** | 6.6 ± 5.2 | 38.0±4.8 |
>
> In the table above, we reported the results across five different seeds with 100 test episodes after 1 million updates.
>
> Regarding LAPO, we could not reproduce the results reported in the paper, which may be due to the difference of the experiment procedure. In the LAPO paper, the methods are evaluated with three random seeds.
> When using the techniques used in LAPO,  the overall performance of V2AE and infoV2AE was better than that of LAPO.
> While LAPO nicely showed how to leverage the latent space for offline RL, LAPO does not utilize the discrete latent space.
> The difference in performance between LAPO and our methods shows the benefit of leveraging the discrete latent variable.
>
> Regarding IQL, we could fairly reproduce the performance reported in the original IQL paper.
> Interestingly, the techniques used in LAPO did not improve the performance of IQL.
> When using the techniques used in LAPO,  the overall performance of V2AE and infoV2AE was better or comparable to IQL.
> This result shows that incorporating the policy structure can achieve the performance achieved by using the state-of-the-art algorithm for the critic.
> We believe that this result supports our aim to shed lights on the importance of the policy structure.
>
> We hope that these additional results address the reviewer's concern. We would appreciate it if the reviewer could let us know the opinion after checking these results.

---

### Official Review · Reviewer_7tms · 2022-10-24

**Confidence:** 4
**Correctness:** 3
**Technical Novelty And Significance:** 3
**Empirical Novelty And Significance:** 3
**Recommendation:** 8

**Clarity, Quality, Novelty And Reproducibility:**

The paper is well written and clear. As far as I can tell, the idea is novel and (mostly) reproducible, modulo the implementation details mentioned above.

**Strength And Weaknesses:**

# Strengths
The paper is well-written and well motivated, and the algorithm is novel (as far as I can tell).

I also wanted to highlight the comparison between the original results of AWAC and the authors in Appendix E (although more to say about this point below).

# Weaknesses
The main weaknesses of this paper are in evaluation.

## Choice of $|\mathcal{Z}|$
The first point is in the choice of the cardinality of $|\mathcal{Z}|$, which is crucial to the algorithm. I feel that the importance of this choice is not highlighted enough (the first time I saw it discussed was in section 7.2 in page 7). In Algorithm 1, for instance, it is never mentioned that $|\mathcal{Z}|$ needs to be selected.

I appreciate that the effect of the dimensionality choice was evaluated to some extent in section 7.2, but I think this is an analysis that needs to be performed more extensively, as it is one of the core components of the proposed method. Some questions that were left unanswered for me:
* How does one pick $|\mathcal{Z}|$?
* How does one know if it'd be better to pick cV2AE instead (where $|\mathcal{Z}|$ doesn't have to be picked)?
* How sensitive is the algorithm to $|\mathcal{Z}|$ in general?

## Implementation details
In section 7.2 the authors say "For TD3+BC and CQL, we used the author-provided implementations." This makes for a comparison I do not trust. Implementation matters and there are often subtle design choices that have a big impact on performance. For proper "apples-to-apples" comparison, you should really be using the same codebase, otherwise the results are suspect. Furthermore, it seems you used your own implementation of TD3+BC for the toy experiments... Why did you not use the same one for the D4RL benchmark tasks?

Indeed, right below table 4 the authors say: "Remarkably, our implementation of AWAC showed significantly better performance than that reported by Nair et al. (2020)." This is precisely why it's not good to compare across implementations.

## Statistical significance
In the tables presented there are bolded numbers even when improvements are not statistically significant: you should only be bolding numbers when there are no overlapping CIs, which is not the case for most of e.g. Table 3 and Table 5.


# Questions / suggestions
1. In equation (2), it seems $\mathcal{Z}$ has to have a pre-specified size. How is that chosen?
1. In equation (5), to the right of the inequality I think it should be $d^{\beta}(\mathbf{s})$
1. In equations (7) and (10) the $\mathbf{z}$ is not qualified. Are you missing a sum or an integral over $\mathcal{Z}$?
1. At the end of equation (10) shouldn't it be $\mathbf{s_i,a_i}$?
1. In Algorithm 1 you should have $\lbrace\ldots \rbrace$ surrounding $(s, a, s', r)$ to indicate its a set of transitions. You should also have a for loop or something like that saying "For each element $(s, a, s', r)$ in the mini-batch...
1. In the second expectation in equation (12), shouldn't it be $a\sim\pi_{z'}$?
1. In the paragraph above equation (10) it should say "In addition, we also propose a regularization" (not an)

**Summary Of The Paper:**

This paper proposes a novel algorithm for offline RL that works by dividing the state-action space into discrete latent regions, learning a deterministic policy for each, and acting with a mixture of these policies. The authors demonstrate the effectiveness of their method on a series of experiments from the D4RL benchmarking datsets and argue that it helps avoid issues with out-of-distribution actions.

**Summary Of The Review:**

This paper presents an interesting and novel algorithm for dealing with out-of-distribution actions. The writing, motivation, and algorithmic description are clear.

My main concern with this paper is in the evaulation of the method, sensitivity analysis of one of its core components, and comparison to baselines (see above for details).

For these reasons I am leaning towards an accept.

---

> ### Author Response · Authors · 2022-11-11
> **Response to Reviewer 7tms**
>
> Thank you for evaluating our paper and giving constructive feedback to us. We summarize our response to the comments and our revisions.
>
> >How does one pick $|Z|$? & How sensitive is the algorithm to $|Z|$  in general?
>
> As long as we used from 8 to 32, we did not clearly see significant differences in performance on D4RL benchmark tasks. In this sense, V2AE is not as sensitive as the option critic in the context of online RL.
> However, if we use too small values for $|Z|$, the policy will not be sufficiently expressive. In addition, if we use larger values for $|Z|$, it will increase the computational complexity. Thus, one can try the value between 8 to 32 when the task complexity is comparable to the D4RL tasks.
>
> To further investigate the sensitivity to $|Z|$, we are running experiments with different values of  $|Z|$ on mujoco-v2 tasks for now.
> We report the result when the experiments are done.
>
> >How does one know if it'd be better to pick cV2AE instead (where |Z| doesn't have to be picked)?
>
> As we showed in the experiment in Figure 3., the learning process of cV2AE could be unstable in some tasks. We think that the cause of the instability is the interpolation of multimodal distribution with the continuous latent variable. Thus, we think that the V2AE is usually the safer choice than cV2AE. However, when it is known that the distribution of the samples in the given dataset is unimodal, one can pick cV2AE.
>
> >Furthermore, it seems you used your own implementation of TD3+BC for the toy experiments... Why did you not use the same one for the D4RL benchmark tasks?
>
> It seems that our description caused some confusion. We used the same implementation for both toy experiments and D4RL benchmark tasks. In other words, we used the author-implementation of TD3+BC for the toy experiments. To clarify this point, we revised the manuscript.
>
> >This is precisely why it's not good to compare across implementations.
>
> We agree with the reviewer. Actually, our implementation of V2AE, AWAC, and easyBCQ is based on the author's implementation of TD3+BC. Thus, our experiments are based on the same code base, and we think that we are doing the proper “apple-to-apple” comparison.
> Regarding the CQL, we could not implement the CQL that achieves a performance comparable to the author's implementation of CQL. As reported by Fujimoto et al. (2021), it is necessary to implement several techniques to achieve the performance reported in the original CQL paper. Thus, we report the result of running the author-implemented CQL in our computation environment. We think this is common practice in the community. We also find several papers that compare the performance of author-implementation of baseline methods from different code bases.
>
> >In the tables presented there are bolded numbers even when improvements are not statistically significant: you should only be bolding numbers when there are no overlapping CIs, which is not the case for most of e.g. Table 3 and Table 5.
>
> As we used the bold numbers to indicate the best performance, we think that it is ok to use the bold numbers for the results where the numbers are currently bold. Regarding the statistical differences, it seems that there are some results that should be bold additionally, e.g. TD3+BC on the walker2d-medium-v2 task. We will fix it in the revised version.
>
> >In equation (5), to the right of the inequality I think it should be $d^{\beta}(s)$
>
> Thank you for pointing out the typo. We will fix it in the revised version.
>
> >In equations (7) and (10) the z is not qualified. Are you missing a sum or an integral over Z?
>
> In the first term of (7), the integral over $z$ is included in the definition of the KL divergence.
> Similarly, the integral over $z$ is included in the expectation for the second term of (7) and (10).
> For clarity, we included the derivation of equation (7) in the appendix.
>
> > At the end of equation (10) shouldn't it be $s_i$,$a_i$?
>
> Thank you for pointing this out. We will fix it in the revised version.
>
> > In Algorithm 1 you should have {…} surrounding (s,a,s′,r)  to indicate its a set of transitions. You should also have a for loop or something like that saying "For each element (s,a,s′,r)  in the mini-batch
>
> Thank you for the comment. We will revise the corresponding part.
>
> > In the second expectation in equation (12), shouldn't it be a∼πz′ ?
>
> Thank you for pointing this out. $s$ and $a$ should be $s’$ and $a’$, respectively.
>
>  > In the paragraph above equation (10) it should say "In addition, we also propose a regularization" (not an)
>
> Thank you for pointing out the typo. We will fix it in the revised version.

---

> > ### Comment · Reviewer_7tms · 2022-11-14
> > **Comparison across implementations**
> >
> > Thank you for your responses, most of them address my concerns.
> >
> > However, I'm still not convinced about the comparison across implementations. Since you stated that "Actually, our implementation of V2AE, AWAC, and easyBCQ is based on the author's implementation of TD3+BC.", could you not run the TD3+BC experiments in your setup to confirm the results obtained?
> >
> > It is not uncommon that code provided with papers is unable to reproduce the paper's results (I have encountered this myself a number of times), which is why it is best to run baselines under the same settings, rather than just copy the numbers from the paper.
> >
> > I realize there are "several papers that compare the performance of author-implementation of baseline methods from different code bases", but this is a poor practice that we shouldn't be continuing.
> >
> > Given that you already have the TD3+BC, could you not run some baselines to verify you can reproduce the paper's results?

---

> > > ### Author Response · Authors · 2022-11-15
> > > **Regarding TD3+BC on mujoco-v2 tasks**
> > >
> > > Thank you for checking our response promptly.
> > >
> > > > Since you stated that "Actually, our implementation of V2AE, AWAC, and easyBCQ is based on the author's implementation of TD3+BC.", could you not run the TD3+BC experiments in your setup to confirm the results obtained?
> > >
> > > We think that the reviewer is asking this because we used the results from the original paper instead of re-running the codes for mujoco-v2 tasks. (We reported the results of re-running the codes for other tasks.)
> > > In our preliminary experiments, we verified that the author's implementation of TD3+BC can reproduce results comparable to those reported in the paper. We reported the results of the original paper for the sake of time. As our experiments are based on the author's implementation of TD3+BC, we initially thought it was ok. However, we understand the intention of the reviewer, and we are now running experiments to evaluate the performance of TD3+BC on mujoco-v2 tasks.
> > > We will report the results of running TD3+BC on mujoco-v2 tasks by ourselves and replace the results reported in the paper.
> > >
> > > We hope that this will address your concern. If you expect any other action, please let us know.

---

> > > > ### Comment · Reviewer_7tms · 2022-11-15
> > > > **Expected action**
> > > >
> > > > This is exactly what i was hoping for, so thank you for running them, and looking forward to seeing the results!

---

> > > > > ### Author Response · Authors · 2022-11-16
> > > > > **Reproducibiliy of TD3+BC**
> > > > >
> > > > > We report the result of re-running TD3+BC on mujoco-v2 tasks as below.
> > > > > Overall, the performance reported in the original TD3+BC paper was fairly reproduced. We will replace the results of TD3+BC in our paper with the results obtained by re-running the authors' implementation.
> > > > >
> > > > > |       | TD3-BC (paper) | TD3-BC (re-run) |
> > > > > | --- | --- | --- |
> > > > > | halfcheetah-expert-v2 | 96.7±1.1 | 96.3±0.9 |
> > > > > | hopper-expert-v2 | 107.8±7.0 | 109.9±2.5 |
> > > > > | walker2d-expert-v2 | 110.2±0.3 | 110.2±0.4 |
> > > > > | halfcheetah-medium-expert-v2 | 90.7±4.3 | 89.4±7.2 |
> > > > > | hopper-medium-expert-v2 | 98.0±9.4 | 95.5±9.4 |
> > > > > | walker2d-medium-expert-v2 | 110.1±0.5 | 110.2±0.3 |
> > > > > | halfcheetah-medium-replay-v2 | 44.6±0.5 | 44.7±0.4 |
> > > > > | hopper-medium-replay-v2 | 60.9±18.8 | 73.8±18.9 |
> > > > > | walker2d-medium-replay-v2 | 81.8±5.5 | 64.5±17.0 |
> > > > > | halfcheetah-medium-v2 | 48.3±0.3 | 48.2±0.3 |
> > > > > | hopper-medium-v2 | 59.3±4.2 | 61.0±4.2 |
> > > > > | walker2d-medium-v2 | 83.7±5.5 | 84.7±1.3 |
> > > > > | halfcheetah-random-v2 | 11.0±1.1 | 11.5±0.6 |
> > > > > | hopper-random-v2 | 8.5±0.6 | 8.7±0.3 |
> > > > > | walker2d-random-v2 | 1.6±1.7 | 1.4±1.9 |
> > > > >
> > > > > We hope that this result resolves the reviewer's concern. If there are any other requests, please let us know.

---

> > > > > > ### Comment · Reviewer_7tms · 2022-11-16
> > > > > > **Thanks!**
> > > > > >
> > > > > > Thank you for running these, and for replacing the results in the paper with those of your re-runs.

---

> ### Author Response · Authors · 2022-11-11
> **Sensitivity to the dimensionality of the latent variable**
>
> As the follow-up, we provide the performance of infoV2AE on walker2d tasks with different $|Z|$.
> As shown, infoV2AE with $|Z|=8$ demonstrated the best performance, while the performance with $|Z|=16$ and $|Z|=32$ is comparable.
> These results show that the performance of the policy is not so sensitive to the dimensionality of the latent varialbe.
> However, the performance with $|Z|=4$ is relatively weak, and it indicates that the policy may not be expressive enough when the dimensionality of the latent varialbe is too small.
>
> |       | infoV2AE  disc=4 | infoV2AE  disc=8 | infoV2AE  disc=16 | infoV2AE  disc=32 |
> | :---: | :---: | :---: | :---: | :---: |
> | walker2d-expert-v2 | 99.7±17.9 | 112.1±0.4 | 108.8±6.8  | 106.4±10.2 |
> | walker2d-medium-expert-v2 | 89.1±25.7 | 110.1±0.7 | 96.0±17.0  | 109.9±0.6 |
> | walker2d-medium-replay-v2 | 81.6±4.5 | 86.7±3.2 | 85.4±3.7 | 86.3±3.1 |
> | walker2d-medium-v2 | 81.8±2.5 | 85.0±0.8 | 69.9±28.3 | 84.3±1.0 |
>
> We will include the above result in the revised manuscript.

---

> ### Comment · Reviewer_7tms · 2022-11-16
> **Adjusting based on rebuttal**
>
> The authors have addressed all my concerns, so I am increasing my score accordingly.

---

### Official Review · Reviewer_RQRe · 2022-10-25

**Confidence:** 3
**Clarity, Quality, Novelty And Reproducibility:** See above.
**Correctness:** 2
**Technical Novelty And Significance:** 3
**Empirical Novelty And Significance:** 2
**Recommendation:** 6

**Strength And Weaknesses:**

Strength:

1. It seems novel to utilize policy structure to improve offline RL algorithms.

2. The paper presents its idea reasonably clearly.

Weaknesses.

I will keep my comments short. I have several critical concerns below.

Motivation. The motivation for using mixture distribution is not persuasive. The paper currently says the offline data may be multimodal, and a unimodal distribution policy can result in OOD actions, as shown in Fig 1. But the mixture distribution could also assign higher density to somewhere between two modes (i.e., Fig 1(c), somewhere between two regions could also have high density). I don’t see a clear benefit of using a mixture distribution.

Lack of insight into the proposed approach. The proposed method has two key components: one is the use of a mixture distribution policy; another one is to use variational lower bound to learn the policy parameters. I expect strong empirical results to show the necessity and effectiveness of the two components. Particularly to show at least the following:

1. The benefit of the proposed algorithm is not due to a more complicated NN. There is a reason to believe that a very large/complicated NN could be beneficial: it is less likely to touch another action’s density when updating one. So probably, the OOD action’s density can be kept low in the learning process. Why not use a mixture of Gaussian as a policy parameterization (with a relatively large NN) and plug it into an offline RL algorithm as a baseline?

2. Why use variational lower bound?

Technical clarification. Eq (3), why such a definition of z? Does it mean pi_gate is also deterministic?

Empirical results:
Why is there no implicit Q learning? IQL empirical works really well and should be included.

**Summary Of The Paper:**

The paper studies an offline RL algorithm by utilizing the special policy structure of the behavior policy (which generates the offline data). The paper proposes to define the policy being learned by using a mixture of sub-policies, which is in the form of pi(s, a) = sum_z p(z|s) pi_sub (a | s, z), where z is the introduced latent variables. The sub policy pi_sub is deterministic and can be written as mu(s, z).  The basic idea behind this form is that the mixture policy is less likely to force the policy to cover all modes in the offline data; covering all modes using a single-mode policy is potentially harmful as it may provide support to many OOD actions, which results in overestimation. The paper then proposes to optimize the policy pi by maximizing the variational lower bound. Experiments are presented on commonly seen testing domains in offline RL literature.

**Summary Of The Review:**

Due to the concerns about the motivation, insight of the proposed method, and empirical results, I tend to reject the paper for now.

---

> ### Author Response · Authors · 2022-11-11
> **Response to Reviewer RQRe (1)**
>
> Thank you for evaluating our paper and giving constructive feedback to us. We summarize our response to the comments and our revisions.
>
> >Why not use a mixture of Gaussian as a policy parameterization (with a relatively large NN) and plug it into an offline RL algorithm as a baseline?
>
> Thank you for the suggestion. We will add the results of running AWAC with a Gaussian mixture policy as an additional baseline if we can make it before the deadline. So far, we obtained the results on Antmaze tasks, which are shown below. As shown, the performance of AWAC with the Gaussian mixture policy is not very different from that of AWAC with a Gaussian policy.
>
> |  |  AWAC(GMM) | AWAC(Gaussian)  | V2AE | infoV2AE |
> | ---- | ---- | ---- |  ---- | ---- |
> |  antmaze-umaze-v0 |  57.4$\pm$6.2 | 49.8$\pm$6.2 | 83.6$\pm$4.5 | 88.4$\pm$3.6 |
> |   antmaze-umaze-diverse-v0  |  46.8$\pm$6.9 | 53.8$\pm$13.0 | 43.2$\pm$ 7.8 | 52.8$\pm$7.9 |
> | antmaze-medium-play-v0 | 0.0$\pm$ 0.0 | 0.0$\pm$0.0 | 77.0$\pm$5.1 | 48.6$\pm$25.3 |
> | antmaze-medium-diverse-v0 | 0.0$\pm$ 0.0 | 0.0$\pm$0.0 | 56.8$\pm$27.2 | 59.2$\pm$29.4 |
> | antmaze-large-play-v0 | 0.0$\pm$ 0.0 | 0.0$\pm$ 0.0 | 1.0$\pm$1.3 | 5.2$\pm$ 8.4 |
> | antmaze-large-diverse-v0 | 0.0$\pm$ 0.0 | 0.0$\pm$ 0.0 | 4.8$\pm$9.6 | 6.6$\pm$ 5.2 |
>
> A recent study by Chen et al. (2022) indicates that the use of the Gaussian mixture policy does not improve the performance of AWAC. When a Gaussian mixture policy is employed, a Gaussian component that covers a large part of the state space often appears,  and it governs the resulting performance. This behavior is also often observed in the context of the option critic for online RL (see Smith et al.,(2018)).  If that happens, we cannot exploit the discrete latent variable. We think that the reviewer is familiar with these observations from previous studies, and we would totally understand it even if the reviewer is skeptical about the GMM policy.
>
> By contrast, we employed a mixture of deterministic policies, not a mixture of the Gaussian policies. Unlike a Gaussian policy,  a deterministic policy can be seen as a distribution given by the dirac-delta function, which is the limit of the Gaussian as the standard deviation goes zero. When we use the mixture of deterministic policies, we will not have a component that covers the large state space.   The use of the mixture of deterministic policies allows us to distribute each component separately, and we can exploit the benefit of the discrete latent variable.
>
> Matthew Smith, Herke Hoof, and Joelle Pineau. An inference-based policy gradient method for
> learning options. In Proceedings of the International Conference on Machine Learning (ICML),
> 2018.
>
> >Why use variational lower bound?
>
> By using the variational lower bound, we can naturally enforce the constraint on the distribution $p(z|s)$ using the KL divergence $D_{KL}(q(z|s,a)||p(z|s))$. This term controls the number of samples assigned to different values of the latent variable $z$. In our case, by using the Boltzman distribution given in equation (8), we assign more samples to the component which is more effective in the region of the state space. This allows us to effectively divide the state-action space during the learning phase.
> We also think that the benefit of using the variational lower bound can be understood from the fact that variational autoencoder is more popular than a simple autoencoder, which is trained by minimizing only the reconstruction error without any constraint on the latent variable.
>
> In addition, we aimed to train the mixture of deterministic policies in this study. Unlike the mixture of Gaussian policies, the objective function for maximum likelihood estimation of the mixture of deterministic policies is not obvious. However, the variational lower bound used in VAE naturally connects the deterministic mapping from the latent space to the sample space. Thus, we used the variational lower bound to derive the objective function for training the mixture of the deterministic policies.
>
> >the mixture distribution could also assign higher density to somewhere between two modes (i.e., Fig 1(c), somewhere between two regions could also have high density).
>
> In Fig1(c), the circles visualize the samples in the given dataset, and the colors indicate the latent variable learned by the proposed method. Thus, the density in Fig1(c) shows the density in the given dataset, not the density induced by running the policy learned by the proposed method.
>
> >Technical clarification. Eq (3), why such a definition of z? Does it mean $\pi_{gate}$ is also deterministic?
>
> We used this gating policy because it achieves optimal behavior when the option policies are given and fixed. As the reviewer understands, $\pi_{gate}$ is deterministic in our implementation. We revised the manuscript to clarify this point.

---

> ### Author Response · Authors · 2022-11-11
> **Response to Reviewer RQRe (2)**
>
> > Empirical results: Why is there no implicit Q learning? IQL empirical works really well and should be included.
>
> We included the comparison with IQL and other baseline methods on mujoco-v2 tasks in the appendix, and V2AE outperformed the IQL on the locomotion tasks in  D4RL. The reason why we excluded IQL from the main text is that the research direction of IQL is orthogonal to ours; IQL is the technique to train the critic, and our focus is the policy structure.  In the main manuscrips, we focused on the baseline methods that use the double-clipped Q-learning in the critic to investigate the effect of the policy structure.
>
> The work of IQL nicely showed that the performance of offline RL can be significantly improved by using the implicit Q-learning for the critic. Our work shows that the performance of offline RL can be improved by considering the policy structure. The fact that V2AE with a simple double-clipped Q-learning outperformed IQL on the locomotion tasks in D4RL shows that incorporating the structured policy can be a complementary approach that can further improve IQL.
>
> In addition, although the performance of V2AE on Antmaze tasks in the current results is lower than that of IQL, we did not use a few techniques to improve the performance for simplicity. We are currently running experiments to evaluate V2AE combined with the techniques used by Chen et al. (2022). We report the results when the experiments are done.
>
> Latent-Variable Advantage-Weighted Policy Optimization for Offline RL Xi Chen, Ali Ghadirzadeh, Tianhe Yu, Yuan Gao, Jianhao Wang, Wenzhe Li, Bin Liang, Chelsea Finn, Chongjie Zhang, NeurIPS 2022.

---

> > ### Author Response · Authors · 2022-11-16
> > **Comparison with IQL on Antmaze**
> >
> > We provide the comparison with IQL on Antmaze tasks as follows.
> > We evaluated IQL by adopting the author's implementation to our code base, and we obtained a performance comparable to that reported in the original IQL paper [1].
> > In the table below, "V2AE" refers to the simple implementation reported in the main manuscript, and "V2AE_lapo" refers to a variant of the proposed method V2AE, which incorporates the techniques used in [2] such as scaling the reward, clipping the target value for the state-value function, adaptive combination of double critics ($Q_{target}=0.7\min(Q_1, Q_2) + 0.3\max(Q_1, Q_2)$).
> >
> > When combined with these techniques, V2AE can achieve a performance comparable to IQL on Antmaze tasks.
> > For a fair comparison, we are also evaluating a variant of IQL which incorporates the techniques in [2]. We will report the results when the experiments are finished.
> > Please also be aware that the performance of IQL is sensitive to implementation techniques such as the cosine scheduling of the learning rate for the actor.
> >
> > We think that the fact that V2AE can achieve a performance comparable to IQL on Antmaze tasks using techniques used in [2] strengthens our contribution.
> > If the reviewer thinks that another experiment is necessary, please let us know.
> >
> > |  | IQL | V2AE | V2AE_lapo |
> > | --- | --- | --- | --- |
> > | antmaze-umaze-v0 | 87.4$\pm$4.5  |  83.6 $\pm$4.5 | **92.8±2.1** |
> > | antmaze-umaze-diverse-v0 | **64.6$\pm$5.6** | 43.2$\pm$ 7.8 | 32.6±25.8 |
> > | antmaze-medium-play-v0 | **74.6$\pm$3.1** | **77.0$\pm$ 5.1** | 63.0±13.0 |
> > | antmaze-medium-diverse-v0 | **73.8$\pm$7.1** | 56.8$\pm$ 27.2 | **75.0±8.5** |
> > | antmaze-large-play-v0 | **39.0$\pm$7.2** | 1.0$\pm$ 1.3 | **42.2±23.0** |
> > | antmaze-large-diverse-v0 | 48.0$\pm$9.0 | 4.8$\pm$ 9.6 | **56.6±4.5** |
> >
> > The table above reports the results across five random seeds with 100 test episodes after 1 million updates.
> >
> > [1] Offline Reinforcement Learning with Implicit Q-Learning. Ilya Kostrikov, Ashvin Nair, Sergey Levine
> > [2] Latent-Variable Advantage-Weighted Policy Optimization for Offline RL. Xi Chen, Ali Ghadirzadeh, Tianhe Yu, Yuan Gao, Jianhao Wang, Wenzhe Li, Bin Liang, Chelsea Finn, Chongjie Zhang, NeurIPS 2022.

---

> > > ### Author Response · Authors · 2022-11-17
> > > **Comparison with IQL**
> > >
> > > As the experiment we promised in the previous comment is now finished, we report the results.
> > > As we mentioned in previous comments, a few techniques to improve performance on Antmaze is incorporated in the authors' implementation of LAPO [1].
> > > As these techniques are not incorporated in our initial experiments, we performed additional experiments for fair comparison.
> > > In methods named like "xx_lapo", the techniques used in LAPO is incorporated.
> > > We summarized the techniques/differences as below:
> > > - computation of the target value as $Q_{target} = 0.7 \min (Q_1, Q_2) + 0.3 \max (Q_1, Q_2)$
> > > - target value clipping $v_{target} =  \min(\max(v_{target}, v_{min}), v_{max})$
> > > - reward scaling $r \leftarrow r\times 100$
> > > - network architecture: three hidden layers with 256 units, learning rate: 2e-4
> > >
> > > In IQL, V2AE, infoV2AE, we used the reward $r \leftarrow r - 1$, following the protocol used in the original IQL paper.
> > >
> > > While action normalization is used in LAPO, we found that the action normalization does not improve the performance of IQL and V2AE.
> > > Therefore, action normalization was not used in IQL_lapo, V2AE_lapo, and infoV2AE_lapo.
> > > "IQL (re-run)" indicates the results of re-running IQL in our code base, which fairly reproduced the results reported in the original IQL paper [2].
> > > "V2AE" and "infoV2AE" show the result reported in our main manuscript.
> > >
> > > |  | IQL(re-run) | IQL_lapo | V2AE | V2AE_lapo | infoV2AE | infoV2AE_lapo |
> > > | --- | --- |--- | --- |--- | --- |--- |
> > > | antmaze-umaze-v0 |  87.4$\pm$4.5  |  87.2$\pm$2.8 | 83.6 $\pm$4.5 | **92.8$\pm$2.1** | 88.4±3.6 | 89.4±5.1 |
> > > | antmaze-umaze-diverse-v0 |  **64.6$\pm$5.6** | 54.0$\pm$13.4 | 43.2$\pm$ 7.8 | 32.6$\pm$25.8 | 52.8±7.9 | 34.8±18.0 |
> > > | antmaze-medium-play-v0 | **74.6$\pm$3.1** | 64.0$\pm$9.5 | **77.0$\pm$ 5.1** | 63.0$\pm$13.0 | 48.6±25.3 | 62.6±6.8 |
> > > | antmaze-medium-diverse-v0 |  73.8$\pm$7.1 | 54.8$\pm$8.6 | 56.8$\pm$ 27.2 | 75.0$\pm$8.5 | 59.2±29.4 | **82.8±4.4** |
> > > | antmaze-large-play-v0 | 39.0$\pm$7.2 | 20.4$\pm$11.0 | 1.0$\pm$ 1.3 | **42.2$\pm$23.0** | 5.2 ± 8.4 | **47.4±14.5** |
> > > | antmaze-large-diverse-v0 | 48.0$\pm$9.0 |  17.0$\pm$4.2 | 4.8$\pm$ 9.6 | **56.6$\pm$4.5** | 6.6 ± 5.2 | 38.0±4.8 |
> > >
> > > In the table above, we reported the results across five different seeds with 100 test episodes after 1 million updates.
> > >
> > > Regarding IQL, we could fairly reproduce the performance reported in the original IQL paper.
> > > Interestingly, the techniques used in LAPO did not improve the performance of IQL, while the performance of V2AE and infoV2AE was improved.
> > > When using the techniques used in LAPO, the overall performance of V2AE and infoV2AE was better or comparable to the original IQL.
> > > This result shows that incorporating the policy structure can achieve the performance achieved by using the state-of-the-art algorithm for the critic.
> > > We believe that this result supports our aim to shed lights on the importance of the policy structure.
> > >
> > > Regarding the comparison with LAPO, we also posted a comment to Reviewer NNys. Please refer to the comment for details.
> > >
> > > [1] Latent-Variable Advantage-Weighted Policy Optimization for Offline RL Xi Chen, Ali Ghadirzadeh, Tianhe Yu, Yuan Gao, Jianhao Wang, Wenzhe Li, Bin Liang, Chelsea Finn, Chongjie Zhang, NeurIPS 2022.
> > > [2] Offline Reinforcement Learning with Implicit Q-Learning. Ilya Kostrikov, Ashvin Nair, Sergey Levine

---

> ### Author Response · Authors · 2022-11-15
> **Additional results on AWAC with a Gaussian mixture policy**
>
> We provide the results on mujoco-v2 tasks as follows:
>
> |      |  mixAWAC | AWAC | V2AE | infoV2AE |
> | --- | --- | --- | --- | --- |
> | halfcheetah-expert-v2 |  94.0±0.5 | 94.8$\pm$ 0.2 | 97.0$\pm$ 1.0 | 95.6$\pm$ 2.0 |
> | hopper-expert-v2 |   111.8±0.8 | 109.8$\pm$ 2.9 | 93.6$\pm$ 15.1 | 107.5$\pm$ 2.9 |
> | walker2d-expert-v2 | 110.5±0.3 | 111.0$\pm$ 0.2 | 111.4$\pm$ 0.3 | 112.1$\pm$ 0.4 |
> | halfcheetah-medium-expert-v2 |  92.1±0.6 |  92.7$\pm$ 0.8 | 91.1$\pm$ 3.4 | 91.4$\pm$ 2.5 |
> | hopper-medium-expert-v2 |  102.0±17.5 | 98.6$\pm$ 10.7 | 78.4$\pm$ 19.0 | 94.5$\pm$14.9 |
> | walker2d-medium-expert-v2 | 109.1±0.3 | 109.2$\pm$ 0.3 | 109.9$\pm$ 0.4 | 110.1$\pm$ 0.7 |
> | halfcheetah-medium-replay-v2 |  41.5±0.4 | 40.9$\pm$ 0.6 | **45.2$\pm$ 0.8** | **46.7$\pm$ 0.6** |
> | hopper-medium-replay-v2 |   41.2±4.7 | 38.2$\pm$ 9.4 | **89.2$\pm$ 8.1** | **98.5$\pm$ 2.0** |
> | walker2d-medium-replay-v2 |   67.7±8.8 | 65.0$\pm$ 15.7 | **82.1$\pm$ 3.8** | **86.7$\pm$ 3.2** |
> | halfcheetah-medium-v2 | 45.1±0.3 | 44.3$\pm$ 0.2 | **47.5$\pm$ 0.4** | **48.6$\pm$ 0.4** |
> | hopper-medium-v2 |  57.2±3.9 |  57.5$\pm$ 3.0 | **71.2$\pm$ 6.5** | **86.4$\pm$ 7.6** |
> | walker2d-medium-v2 | 78.7±4.8 |  81.0$\pm$ 2.5 | 79.4$\pm$ 4.7 | 85.0$\pm$ 0.8 |
> | halfcheetah-random-v2 |  2.2±0.0 | 3.2$\pm$ 1.3 | **15.8$\pm$ 1.6** | **16.3$\pm$ 1.2** |
> | hopper-random-v2 | 8.2±0.2 | 7.3$\pm$ 0.9 | **12.0$\pm$ 10.0** | **20.4$\pm$ 9.8** |
> | walker2d-random-v2 | 4.9±1.1 |  3.1$\pm$ 1.0 | 2.5$\pm$ 2.6 | 2.3$\pm$ 2.0 |
>
> (Bold numbers indicate the results where our methods clearly outperformed AWAC and mixAWAC.)
>
> Overall, the use of the Gaussian mixture policy does not improve the performance of AWAC. In our method, a mixture of deterministic policies is used, which enables us to avoid getting one component that covers the large state space. In addition, we also leverage the variational lower bound to incorporate the prior distribution of the latent variable, which enables us to obtain meaningful latent representations.  These are the reasons why our methods clearly outperformed AWAC, while simply using the Gaussian mixture policy does not improve the performance of AWAC.

---

### Author Response · Authors · 2022-11-11
**Update log**

- Add the derivation of (7) in the appendix
- Revise equation (5),(7),(10),(12)
- Revise Algorithm 1
- Add detailed descriptions about the update of networks in the appendix
- Revise the description of the implementations

---

> ### Author Response · Authors · 2022-11-15
> **Update log 2**
>
> - Add the results of AWAC with a Gaussian mixture policy in the appendix
> - Add the results regarding the sensitivity to the deminsionality of the latent variable in the appendix

---

> ### Author Response · Authors · 2022-11-17
> **Update log 3**
>
> - Replaced the performance of TD3+BC on mujoco-v2 tasks with the results obtained by re-running the authors' implementation.
> - Add the comparison with LAPO in the appendix

---

> ### Author Response · Authors · 2022-11-18
> **Update log 4**
>
> -  Add comparison with LAPO and IQL on Antmaze tasks in the appendix

---

### Decision · Program_Chairs · 2023-01-20

**Decision:**

Reject

**Justification For Why Not Higher Score:**

The specific idea - the mixture of deterministic policies - is not well motivated over other mixture models. Why and when should we pursue this architecture over other mixture models? Is the benefit specific to the dataset used in the experiments or does it translated to other datasets and domains? These questions are quite important to be addressed in the main body of the paper.

**Justification For Why Not Lower Score:**

N/A

**Metareview: Summary, Strengths And Weaknesses:**

This paper proposes using a mixture model for the target policy in offline reinforcement learning. There are similar ideas in the literature, including LAPO presented at NeurIPS this year. The main difference is that this work proposes using the mixture of deterministic policies. The main body of the paper needs to be rewritten to more thoroughly cover the related work for motivate and appreciate the proposed method, specifically a detailed comparison and discussion on why we should better choose the mixture of deterministic policies over other mixture policies. The lack of comparison to related work is undermining the novelty of the paper.

Related to the above point, it seems that the actual benefit might be coming from the reduction of the variance using determinisic policies, yet making them more expressive by taking the mixture of them. If so, this would also require rewriting the narrative in the paper.


**Summary Of Ac-Reviewer Meeting:**

There was disagreement among the reviewers on the novelty aspect about the paper. Although the idea itself - the mixture of deterministic policies - hasn't been proposed in the literature, the mixture (of stochastic) policy has been proposed in a number of papers with variations, many of them with principled motivation. In that sense, the paper fell short in fully motivating the main idea of the paper and one of the reviewers evaluated it "ad-hoc" for the specific datasets used for experiments.

As stated in my meta-review, the actual benefit might be coming from the variance reduction, but the paper in its current form doesn't clearly show why we want to choose the mixture of deterministic policies over other mixture models.